# Online Bayesian Experimental Design for Partially Observed Dynamical Systems

**Sara Pérez-Vieites** [1]  **Sahel Iqbal** [2]  **Simo Särkkä** [3][4]  **Dominik Baumann** [4]

## Abstract

Bayesian experimental design (BED) provides a principled framework for optimising data collection by choosing experiments that are maximally informative about unknown parameters. However, existing methods cannot deal with the joint challenge of (a) *partially observable dynamical systems*, where only noisy and incomplete observations are available, and (b) *fully online inference*, which updates posterior distributions and selects designs sequentially in a computationally efficient manner. Under partial observability, dynamical systems are naturally modeled as state-space models (SSMs), in which latent states mediate the link between parameters and data, making the likelihood—and thus information-theoretic objectives like the expected information gain (EIG)—intractable. We address these challenges by deriving new estimators of the EIG and its gradient that explicitly marginalise latent states, enabling scalable stochastic optimisation in nonlinear SSMs. Our approach leverages nested particle filters for efficient online state-parameter inference with convergence guarantees. Applications to realistic models, such as the susceptible–infectious–recovered (SIR) model and a moving source location task, show that our method successfully handles both partial observability and online inference.

## 1. Introduction

Designing experiments that yield maximally informative data is a fundamental problem across science and engineering. The ability to guide data collection efficiently can accelerate learning in diverse domains, including material discovery (Lei et al., 2021; Lookman et al., 2019), DNA sequencing (Weilguny et al., 2023), sensor networks (Wu et al., 2023), and drug discovery (Lyu et al., 2019). Formally, given a known model structure, let $\boldsymbol{\theta}$ represent the parameters of interest and $\boldsymbol{\xi}$ the design chosen for the experiment, which determines the corresponding observation $\boldsymbol{y}$. The objective is to select designs that maximise the information gained about $\boldsymbol{\theta}$, thereby reducing uncertainty as more data becomes available.

Bayesian experimental design (BED) provides a principled framework for combining Bayesian inference with data acquisition (Lindley, 1956; Huan et al., 2024). In this setting, prior beliefs about the parameters $p(\boldsymbol{\theta})$ are updated as data are collected, yielding a posterior distribution that quantifies uncertainty. This update is driven by the likelihood $p(\boldsymbol{y} \mid \boldsymbol{\theta}, \boldsymbol{\xi})$, which links parameters to observations under a given design $\boldsymbol{\xi}$. When this process is repeated adaptively, using past observations to guide future design choices, we obtain the sequential BED setting (Rainforth et al., 2024). At the experiment $t$, the prior is $p(\boldsymbol{\theta} \mid h_{t-1})$, where $h_{t-1} = \{\boldsymbol{\xi}_{1:t-1}, \boldsymbol{y}_{1:t-1}\}$ is the history of past designs and observations. The next design $\boldsymbol{\xi}_t$ is then chosen to maximise some utility function, commonly the expected information gain (EIG), which requires evaluating $p(\boldsymbol{y}_t \mid \boldsymbol{\theta}, \boldsymbol{\xi}_t)$.

A common assumption in sequential BED is that the likelihood $p(\boldsymbol{y}_t \mid \boldsymbol{\theta}, \boldsymbol{\xi}_t)$ is tractable and available in closed form. While this holds in some settings, it is unrealistic in many real-world scenarios where the system cannot be fully observed. In particular, dynamical systems often involve latent states that evolve over time and are only observed through noisy, partial measurements. Such systems are naturally modeled as state-space models (SSMs) (Särkkä & Svensson, 2023), where the likelihood is defined only indirectly through integration over the hidden states, and is therefore *intractable*.

Several BED approaches address *implicit* likelihoods, i.e., when we can simulate observations from the model but cannot evaluate $p(\boldsymbol{y}_t \mid \boldsymbol{\theta}, \boldsymbol{\xi}_t)$ in closed form. In principle, such methods are relevant to our problem setting, since SSMs allow forward simulation of latent trajectories and observations. However, a direct application would not be straightforward, since the BED objective depends on marginal like-

[1]Department of Computer Science, University of Helsinki, Finland [2]Department of Statistics, University of Oxford, UK [3]ELLIS Institute Finland, Finland [4]Department of Electrical Engineering and Automation, Aalto University, Finland. Correspondence to: Sara Pérez-Vieites <sara.perezvieites@helsinki.fi>.

*Proceedings of the 43$^{rd}$ International Conference on Machine Learning*, Seoul, South Korea. PMLR 306, 2026. Copyright 2026 by the author(s).

lihoods obtained by integrating over latent states, and sequential decisions require history-dependent belief updates. Moreover, as the horizon grows, the space of possible latent trajectories and observation histories expands, making coverage challenging for offline-trained surrogates. Accordingly, implicit-likelihood BED methods have been proposed mostly in static settings (Kleinegesse & Gutmann, 2020; Dehideniya et al., 2018), with recent sequential extensions relying on *amortisation* (Ivanova et al., 2021; Kleinegesse & Gutmann, 2021). In these approaches, surrogate components (e.g., a neural network) are trained offline on simulated data and then reused at deployment, for example to approximate likelihood ratios, mutual-information bounds, or design policies. We discuss these connections in more detail in Section 4, clarifying the differences in problem setting and operational constraints.

In contrast, we focus on a *non-amortised, fully online* regime: designs are optimised sequentially using the current posterior and without offline training. This regime is motivated by applications where (i) decisions are made over long horizons, (ii) the system is only partially observed, and (iii) computational budget between measurements can vary—often prioritizing reliability over ultra-fast deployment, e.g., adaptive clinical trials (Chaloner & Verdinelli, 1995) and astronomical observations (Loredo, 2004). A key practical issue in such settings is *long-horizon feasibility*. In partially observed dynamical models, robust inference/design schemes that repeatedly rely on the full data history and long latent trajectories can incur computation and memory costs that grow with $T$, which may become prohibitive in practice. Our aim is therefore to develop methods that keep inference and design updates *online* (with constant per-step computational cost and memory in the horizon length) while retaining convergence guarantees for the estimators used in design optimisation. We discuss these trade-offs and connections to prior work in Section 4.

**Contributions.** In this paper, we propose a novel method for sequential BED in partially observable dynamical systems, where likelihoods are intractable due to latent-state marginalisation and online inference is required. Our main contributions are (i) *new estimators* of EIG and its gradient, specifically derived for state-space models with latent states, enabling design optimisation under partial observability; (ii) an *online* sequential BED approach that couples these estimators with nested particle filters (NPFs, Crisan & Míguez, 2018), which recursively updates the joint parameter–state posterior without reprocessing the full history, yielding inference and design updates with linear (constant) cost in the horizon; and (iii) *asymptotic consistency guarantees* for the resulting EIG estimators when implemented with our approach.

## 2. Background

### 2.1. Sequential Bayesian Experimental Design

The goal of sequential BED is to select, at each experiment $t$, a design $\boldsymbol{\xi}_t$ that maximises the expected information gain (EIG, Lindley, 1956) about the parameters: $\boldsymbol{\xi}_t^\star = \arg\max_{\boldsymbol{\xi}_t \in \Omega} \mathcal{I}(\boldsymbol{\xi}_t)$, where $\Omega$ is the design space. The EIG quantifies the reduction in posterior uncertainty about $\boldsymbol{\theta}$ after observing $\boldsymbol{y}_t$ under design $\boldsymbol{\xi}_t$,

$$\mathcal{I}(\boldsymbol{\xi}_t) = \mathbb{E}_{p(\boldsymbol{y}_t|\boldsymbol{\xi}_t)}\bigg[ \mathcal{H}[p(\boldsymbol{\theta}\,|\,h_{t-1})] - \mathcal{H}[p(\boldsymbol{\theta}\,|\,h_t)]\bigg] \qquad (1)$$

$$= \mathbb{E}_{p(\boldsymbol{\theta}|h_{t-1})\,p(\boldsymbol{y}_t|\boldsymbol{\theta},\boldsymbol{\xi}_t)}\big[ \log p(\boldsymbol{y}_t\,|\,\boldsymbol{\theta},\boldsymbol{\xi}_t) - \log p(\boldsymbol{y}_t\,|\,\boldsymbol{\xi}_t)\big],$$

with $h_{t-1} = \{\boldsymbol{\xi}_{1:t-1}, \boldsymbol{y}_{1:t-1}\}$ the history, $p(\boldsymbol{y}_t\,|\,\boldsymbol{\theta},\boldsymbol{\xi}_t)$ the likelihood, $p(\boldsymbol{y}_t\,|\,\boldsymbol{\xi}_t)$ the marginal predictive distribution, and $\mathcal{H}$ denoting entropy.

In many applications, however, observations $\boldsymbol{y}_t$ are not generated directly by $\boldsymbol{\theta}$, but through an underlying dynamical system with latent states $\boldsymbol{x}_{0:t}$. Such cases are naturally modeled as SSMs, for which the likelihood is not available in closed form, rendering standard BED methods inapplicable.

### 2.2. State-Space Models (SSMs)

We focus on dynamical systems that can be described by Markovian SSMs. At each experiment or time step $t \in \mathbb{N}$, the system state $\boldsymbol{x}_t \in \mathbb{R}^{d_x}$ evolves and produces an observation $\boldsymbol{y}_t \in \mathbb{R}^{d_y}$:

$$\boldsymbol{x}_t \quad \sim \quad f(\boldsymbol{x}_t\,|\,\boldsymbol{x}_{t-1},\boldsymbol{\theta},\boldsymbol{\xi}_t), \qquad (2)$$

$$\boldsymbol{y}_t \quad \sim \quad g(\boldsymbol{y}_t\,|\,\boldsymbol{x}_t,\boldsymbol{\theta},\boldsymbol{\xi}_t), \qquad (3)$$

where $\boldsymbol{\theta} \in \mathbb{R}^{d_\theta}$ are *unknown* parameters and $\boldsymbol{\xi}_t \in \Omega \subset \mathbb{R}^{d_\xi}$ is the design variable. Here, $f(\boldsymbol{x}_t\,|\,\boldsymbol{x}_{t-1},\boldsymbol{\theta},\boldsymbol{\xi}_t)$ is the transition density of the latent states, and $g(\boldsymbol{y}_t\,|\,\boldsymbol{x}_t,\boldsymbol{\theta},\boldsymbol{\xi}_t)$ is the observation model, both assumed to be differentiable with respect to (w.r.t.) $\boldsymbol{\xi}_t$. We assume that $\boldsymbol{y}_t$ (conditional on the state, parameters, and design) is conditionally independent of all other observations, and the prior probability density functions (pdfs) of the state and the parameters, $p(\boldsymbol{\theta})$ and $p(\boldsymbol{x}_0)$, are known and independent.

### 2.3. Particle Filters (PFs)

In SSMs, the likelihood $p(\boldsymbol{y}_t\,|\,\boldsymbol{\theta},\boldsymbol{\xi}_t,h_{t-1})$ typically has no closed form, as it requires marginalising the latent state,

$$p(\boldsymbol{y}_t\,|\,\boldsymbol{\theta},\boldsymbol{\xi}_t,h_{t-1}) = \int g(\boldsymbol{y}_t\,|\,\boldsymbol{x}_t,\boldsymbol{\theta},\boldsymbol{\xi}_t)\,p(\mathrm{d}\boldsymbol{x}_t\,|\,\boldsymbol{y}_{1:t},\boldsymbol{\theta},\boldsymbol{\xi}_{1:t}). \qquad (4)$$

Particle filters (PFs, Gordon et al., 1993; Doucet et al., 2000; 2001; Djurić et al., 2003) provide a standard solution by recursively approximating the filtering or posterior distribu-

tion with a weighted set of $M$ particles,

$$p(\mathrm{d}\boldsymbol{x}_t \,|\, \boldsymbol{y}_{1:t}, \boldsymbol{\theta}, \boldsymbol{\xi}_{1:t}) \approx \sum_{m=1}^{M} w_{\boldsymbol{x},t}^{(m)} \, \delta_{\boldsymbol{x}_t^{(m)}}(\mathrm{d}\boldsymbol{x}_t).$$

At each time step, particles $\boldsymbol{x}_t^{(m)}$ are propagated forward using the state transition $f$, while their weights $w_{\boldsymbol{x},t}^{(m)}$ are updated in proportion to the observation model $g$. This allows the algorithm to adaptively focus on regions of high likelihood and provides a flexible representation of complex, nonlinear distributions.

Standard PFs assume that both the parameters $\boldsymbol{\theta}$ and the designs $\boldsymbol{\xi}_{1:t}$ are known and fixed, so that only the latent states need to be inferred. In sequential BED, however, the parameters $\boldsymbol{\theta}$ are also unknown and must be estimated. This motivates the use of NPFs, which extend the PF framework to incorporate parameter inference.

### 2.4. Nested Particle Filters (NPFs)

NPFs (Crisan & Míguez, 2018) extend PFs to jointly estimate states and parameters *online*, targeting the joint posterior $p(\mathrm{d}\boldsymbol{x}_{0:t}, \mathrm{d}\boldsymbol{\theta} \,|\, \boldsymbol{y}_{1:t}, \boldsymbol{\xi}_{1:t})$. An NPF employs two intertwined layers of PFs: (i) an *inner* layer that estimates the state distribution conditional on each parameter particle

$$p(\mathrm{d}\boldsymbol{x}_t \,|\, \boldsymbol{y}_{1:t}, \boldsymbol{\theta}, \boldsymbol{\xi}_{1:t}) \approx \sum_{m=1}^{M} \sum_{n=1}^{N} w_{\boldsymbol{\theta},t}^{(m)} w_{\boldsymbol{x},t}^{(m,n)} \, \delta_{\boldsymbol{x}_t^{(m,n)}}(\mathrm{d}\boldsymbol{x}_t),$$

and (ii) an *outer* layer that represents the parameter posterior

$$p(\mathrm{d}\boldsymbol{\theta} \,|\, \boldsymbol{y}_{1:t}, \boldsymbol{\xi}_{1:t}) \approx \sum_{m=1}^{M} w_{\boldsymbol{\theta},t}^{(m)} \, \delta_{\boldsymbol{\theta}_t^{(m)}}(\mathrm{d}\boldsymbol{\theta}).$$

The parameter weights, $w_{\boldsymbol{\theta},t}^{(m)}$, are updated using an estimate of the likelihood in (4) obtained from the inner filters.

This nested structure is related to sequential Monte Carlo squared (SMC$^2$, Chopin et al., 2013), but differs in a key aspect. In SMC$^2$, when parameters are moved (e.g., via MCMC rejuvenation), the inner filters are typically re-run using the full data history, leading to a cost that grows quadratically with time. This becomes problematic in long-horizon settings, both in runtime and memory. NPFs instead use a *jittering* kernel to perturb parameter particles,

$$\boldsymbol{\theta}_t^{(m)} \sim \kappa_M\big(\mathrm{d}\boldsymbol{\theta} \,|\, \boldsymbol{\theta}_{t-1}^{(m)}\big),$$

with a variance chosen to satisfy the regularity conditions required for the NPF convergence results (Crisan & Míguez, 2018), and tuned in practice to balance exploration and stability. This controlled perturbation enables a purely online update: each inner PF is advanced a *single* step (rather than reprocessing past data), yielding linear cost in $t$.

Several extensions replace the inner PF with more structured approximations (e.g., Gaussian or Rao–Blackwellised variants) while retaining theoretical guarantees under modified assumptions (Pérez-Vieites & Míguez, 2021; Fang et al., 2023).

## 3. Online BED for SSMs

We propose a new sequential BED method for partially observable dynamical systems, deriving recursive estimators of the EIG and its gradient. This enables online optimisation in continuous design spaces, $\boldsymbol{\xi}_t \in \mathbb{R}^{d_\xi}$, using stochastic gradient ascent (SGA). We further establish the consistency of the proposed EIG estimator, and conclude the section by presenting the overall algorithmic scheme.

### 3.1. EIG in SSMs

The EIG expression in (1) shows the dependency on the likelihood $p(\boldsymbol{y}_t \,|\, \boldsymbol{\theta}, \boldsymbol{\xi}_t)$ and the marginal likelihood (evidence) $p(\boldsymbol{y}_t \,|\, \boldsymbol{\xi}_t) = \mathbb{E}_{p(\boldsymbol{\theta}|h_{t-1})}[p(\boldsymbol{y}_t \,|\, \boldsymbol{\theta}, \boldsymbol{\xi}_t)]$. Even when the likelihood is available in closed-form, estimating $\mathcal{I}(\boldsymbol{\xi}_t)$ is *doubly intractable* because the evidence appears inside the outer expectation that defines the utility, typically needing nested Monte Carlo (NMC) methods or other approximations (Rainforth et al., 2018; Foster et al., 2019).

In partially observable dynamical systems, the difficulty compounds because both likelihood and evidence require marginalisation over latent states. For compactness, we denote the likelihood and the evidence as

$$L_{\boldsymbol{\theta}, \boldsymbol{\xi}_t}(\boldsymbol{y}_t) \coloneqq \mathbb{E}_{p(\boldsymbol{x}_{0:t}|\boldsymbol{\theta}, h_{t-1})}\big[g(\boldsymbol{y}_t \,|\, \boldsymbol{x}_t, \boldsymbol{\theta}, \boldsymbol{\xi}_t)\big],$$
$$Z_{\boldsymbol{\xi}_t}(\boldsymbol{y}_t) \coloneqq \mathbb{E}_{p(\boldsymbol{\theta}|h_{t-1})\, p(\boldsymbol{x}_{0:t}|\boldsymbol{\theta}, h_{t-1})}\big[g(\boldsymbol{y}_t \,|\, \boldsymbol{x}_t, \boldsymbol{\theta}, \boldsymbol{\xi}_t)\big],$$

respectively. Substituting these into (1) yields

$$\mathcal{I}(\boldsymbol{\xi}_t) = \mathbb{E}_{p(\boldsymbol{\theta}|h_{t-1})\, p(\boldsymbol{y}_t, \boldsymbol{x}_{0:t}|\boldsymbol{\theta}, \boldsymbol{\xi}_t)} \left[ \log \frac{L_{\boldsymbol{\theta}, \boldsymbol{\xi}_t}(\boldsymbol{y}_t)}{Z_{\boldsymbol{\xi}_t}(\boldsymbol{y}_t)} \right]. \quad (5)$$

This expression makes explicit that evaluating the EIG requires not only the usual outer expectation over $\boldsymbol{y}_t$ (and $\boldsymbol{\theta}$), but also inner expectations over latent-state trajectories inside both $L_{\boldsymbol{\theta}, \boldsymbol{\xi}_t}$ and $Z_{\boldsymbol{\xi}_t}$. Although these quantities are analytically intractable, they admit natural Monte Carlo approximations: $L_{\boldsymbol{\theta}, \boldsymbol{\xi}_t}(\boldsymbol{y}_t)$ can be estimated by sampling state trajectories conditional on a fixed $\boldsymbol{\theta}$, while $Z_{\boldsymbol{\xi}_t}(\boldsymbol{y}_t)$ requires sampling jointly from the parameter prior and state dynamics. This compounded intractability motivates the gradient representation and efficient Monte Carlo estimators that we develop next.

### 3.2. Gradient of the EIG

Using Fisher's identity (Douc et al., 2014) on the EIG of (5), we obtain a gradient representation that separates (i)

derivatives of the likelihood and evidence and (ii) the design score at time $t$. Let

$$\Gamma_{\boldsymbol{\xi}_t}(\boldsymbol{y}_t, \boldsymbol{x}_{0:t}, \boldsymbol{\theta}) := g(\boldsymbol{y}_t \,|\, \boldsymbol{x}_t, \boldsymbol{\theta}, \boldsymbol{\xi}_t) f(\boldsymbol{x}_t \,|\, \boldsymbol{x}_{t-1}, \boldsymbol{\theta}, \boldsymbol{\xi}_t) \times$$
$$\times \, p(\boldsymbol{x}_{0:t-1} \,|\, \boldsymbol{\theta}, h_{t-1}) p(\boldsymbol{\theta} \,|\, h_{t-1})$$

be the joint distribution of states, parameters, and observations given a design (generative model at time $t$). We assume the availability of an approximation to the joint posterior distribution of states and parameters at the previous time step, $p(\boldsymbol{x}_{0:t-1}, \boldsymbol{\theta} \,|\, h_{t-1})$.

The gradient of the EIG can be written as

$$\nabla_{\boldsymbol{\xi}_t} \mathcal{I}(\boldsymbol{\xi}_t) = \mathbb{E}_{\Gamma_{\boldsymbol{\xi}_t}} \Bigg[ \frac{\nabla_{\boldsymbol{\xi}_t} L_{\boldsymbol{\theta}, \boldsymbol{\xi}_t}(\boldsymbol{y}_t)}{L_{\boldsymbol{\theta}, \boldsymbol{\xi}_t}(\boldsymbol{y}_t)} - \frac{\nabla_{\boldsymbol{\xi}_t} Z_{\boldsymbol{\xi}_t}(\boldsymbol{y}_t)}{Z_{\boldsymbol{\xi}_t}(\boldsymbol{y}_t)}$$
$$+ \log \left[ \frac{L_{\boldsymbol{\theta}, \boldsymbol{\xi}_t}(\boldsymbol{y}_t)}{Z_{\boldsymbol{\xi}_t}(\boldsymbol{y}_t)} \right] s_{\boldsymbol{\theta}, \boldsymbol{\xi}_t}(\boldsymbol{x}_{t-1:t}, \boldsymbol{y}_t) \Bigg], \quad (6)$$

where $\Gamma_{\boldsymbol{\xi}_t}$ stands for $\Gamma_{\boldsymbol{\xi}_t}(\boldsymbol{y}_t, \boldsymbol{x}_{0:t}, \boldsymbol{\theta})$, and the design score at time $t$ is

$$s_{\boldsymbol{\theta}, \boldsymbol{\xi}_t}(\boldsymbol{x}_{t-1:t}, \boldsymbol{y}_t) := \nabla_{\boldsymbol{\xi}_t} \log g(\boldsymbol{y}_t \,|\, \boldsymbol{x}_t, \boldsymbol{\theta}, \boldsymbol{\xi}_t)$$
$$+ \nabla_{\boldsymbol{\xi}_t} \log f(\boldsymbol{x}_t \,|\, \boldsymbol{x}_{t-1}, \boldsymbol{\theta}, \boldsymbol{\xi}_t),$$

with any of the terms vanishing when they do not depend on $\boldsymbol{\xi}_t$. See details in Appendix A.

Direct evaluation of (6) is intractable. Both the likelihood $L_{\boldsymbol{\theta}, \boldsymbol{x}_t}(\boldsymbol{y}_t)$ and the evidence $Z_{\boldsymbol{x}_t}(\boldsymbol{y}_t)$, as well as their gradients, are defined by expectations over latent-state trajectories. For instance, to sample a new state $\boldsymbol{x}_t$ given a new $\boldsymbol{\theta}$, one needs the full path

$$p(\boldsymbol{x}_{0:t} \,|\, \boldsymbol{\theta}, h_{t-1})$$
$$\propto p(\boldsymbol{x}_0) \prod_{s=1}^{t} f(\boldsymbol{x}_s \,|\, \boldsymbol{x}_{s-1}, \boldsymbol{\theta}, \boldsymbol{\xi}_s) \prod_{s=1}^{t-1} g(\boldsymbol{y}_s \,|\, \boldsymbol{x}_s, \boldsymbol{\theta}, \boldsymbol{\xi}_s),$$

which cannot be updated incrementally without revisiting all $s = 1, \ldots, t-1$. Naïvely, this yields $\mathcal{O}(t^2)$ cost across $t$ steps, as each new gradient evaluation requires resimulating and reweighting entire trajectories.

In the next subsection, we show how this quadratic bottleneck can be avoided using NPFs, yielding Monte Carlo gradient estimators that require only a *single* forward propagation step per iteration, i.e., with complexity $\mathcal{O}(t)$.

### 3.3. Monte Carlo Estimators using NPF

We now instantiate the terms in (6) using the approximation of the posterior distributions available at time $t-1$, obtained with the NPF framework. Let $\{\boldsymbol{\theta}_{t-1}^{(m)}, w_{\boldsymbol{\theta}, t-1}^{(m)}\}_{m=1}^{M}$ denote the outer (parameter) particles and, for each $m$, let $\{\boldsymbol{x}_{0:t-1}^{(m,n)}, w_{\boldsymbol{x}, t-1}^{(m,n)}\}_{n=1}^{N}$ be the inner (state-trajectory) particles. This yields particle approximations to $p(\boldsymbol{\theta} \,|\, h_{t-1})$ and

$p(\boldsymbol{x}_{0:t-1} \,|\, \boldsymbol{\theta}, h_{t-1})$ that we *update online* without reprocessing past data.

For approximating the EIG or any other quantity in (5) or (6) given a design $\boldsymbol{\xi}_t$, we first sample from the joint $\Gamma_{\boldsymbol{\xi}_t}(\boldsymbol{y}_t, \boldsymbol{x}_{0:t}, \boldsymbol{\theta})$. In the NPF framework, we reuse samples from the previous posterior and only draw one-step predictive states and pseudo-observations

$$\widetilde{\boldsymbol{x}}_t^{(m,n)} \sim f(\boldsymbol{x}_t \,|\, \boldsymbol{x}_{t-1}^{(m,n)}, \boldsymbol{\theta}_{t-1}^{(m)}, \boldsymbol{\xi}_t),$$
$$\widetilde{\boldsymbol{y}}_t^{(m,n)} \sim g(\boldsymbol{y}_t \,|\, \widetilde{\boldsymbol{x}}_t^{(m,n)}, \boldsymbol{\theta}_{t-1}^{(m)}, \boldsymbol{\xi}_t).$$

Thus, Monte Carlo samples from $\Gamma_{\boldsymbol{\xi}_t}$ form the array $\{(\boldsymbol{\theta}_{t-1}^{(m)}, \boldsymbol{x}_{0:t-1}^{(m,n)}, \widetilde{\boldsymbol{x}}_t^{(m,n)}, \widetilde{\boldsymbol{y}}_t^{(m,n)})\}$, needed for the *outer* expectation in (6), with weights

$$w_{\boldsymbol{y}, t}^{(m,n)} = w_{\boldsymbol{\theta}, t-1}^{(m)} \, w_{\boldsymbol{x}, t-1}^{(m,n)}.$$

For notational clarity, we relabel the pseudo-observations as $\widetilde{\boldsymbol{y}}_t^{(\ell)}$, $\ell = 1, \ldots, L$, $L = M \times N$, with corresponding weights $w_{\boldsymbol{y}, t}^{(\ell)}$, where each index $\ell$ corresponds to a unique pair $(m, n)$.

The samples required for the *inner* expectations that integrate out latent states are sampled in a similar way. We then sample the next state in the trajectory with different conditioning: (i) for $L_{\boldsymbol{\theta}, \boldsymbol{\xi}_t}(\widetilde{\boldsymbol{y}}_t)$ we *fix* the parameters, $\boldsymbol{\theta}_{t-1}^{(m)}$; and (ii) for $Z_{\boldsymbol{\xi}_t}(\boldsymbol{y}_t)$ we sample new parameters in order to draw new states. With the NPF particles, we can operate:

*(a) For $L_{\boldsymbol{\theta}, \boldsymbol{\xi}_t}(\boldsymbol{y}_t)$:* fix $\boldsymbol{\theta} = \boldsymbol{\theta}_{t-1}^{m}$ and propagate the associated inner particles one step,

$$\ddot{\boldsymbol{x}}_t^{(m,j)} \sim f(\boldsymbol{x}_t \,|\, \boldsymbol{x}_{t-1}^{(m,j)}, \boldsymbol{\theta}_{t-1}^{(m)}, \boldsymbol{\xi}_t),$$

to obtain the Monte Carlo approximation

$$\widehat{L}_{\boldsymbol{\theta}_{t-1}^{(m)}, \boldsymbol{\xi}_t}^N(\boldsymbol{y}_t) = \sum_{j=1}^{N} w_{\boldsymbol{x}, t-1}^{(m,j)} g(\boldsymbol{y}_t \,|\, \ddot{\boldsymbol{x}}_t^{(m,j)}, \boldsymbol{\theta}_{t-1}^{(m)}, \boldsymbol{\xi}_t). \quad (7)$$

*(b) For $Z_{\boldsymbol{\xi}_t}(\boldsymbol{y}_t)$:* jitter parameter particles and propagate one step,

$$\dot{\boldsymbol{\theta}}_{t-1}^{(i)} \sim \kappa_M(\cdot \,|\, \boldsymbol{\theta}_{t-1}^{(i)}),$$
$$\dot{\boldsymbol{x}}_t^{(i,j)} \sim f(\boldsymbol{x}_t \,|\, \boldsymbol{x}_{t-1}^{(i,j)}, \dot{\boldsymbol{\theta}}_{t-1}^{(i)}, \boldsymbol{\xi}_t),$$

then average across both indices with the outer and inner weights,

$$\widehat{Z}_{\boldsymbol{\xi}_t}^{M,N}(\boldsymbol{y}_t) = \sum_{i=1}^{M} \sum_{j=1}^{N} w_{\boldsymbol{\theta}, t-1}^{(i)} w_{\boldsymbol{x}, t-1}^{(i,j)} g(\boldsymbol{y}_t \,|\, \dot{\boldsymbol{x}}_t^{(i,j)}, \dot{\boldsymbol{\theta}}_{t-1}^{(i)}, \boldsymbol{\xi}_t).$$
$$(8)$$

Thus, to approximate the EIG in (5) we draw: (i) parameters $\{\boldsymbol{\theta}_t^{(m)}, \dot{\boldsymbol{\theta}}_t^{(m)}\}$ for $m = 1, \ldots, M$; (ii) conditional state trajectories $\{\boldsymbol{x}_{0:t-1}^{(m,n)}, \widetilde{\boldsymbol{x}}_t^{(m,n)}, \dot{\boldsymbol{x}}_t^{(m,n)}, \ddot{\boldsymbol{x}}_t^{(m,n)}\}$

for $n = 1, \ldots, N$; and (iii) pseudo-observations $\widetilde{\boldsymbol{y}}_t^{(\ell)} \sim g(\boldsymbol{y}_t \,|\, \widetilde{\boldsymbol{x}}_t^{(m,n)}, \boldsymbol{\theta}_t^{(m)}, \boldsymbol{\xi}_t)$, for $\ell = 1, \ldots, L$.[1] This yields the NMC approximation

$$\widehat{\mathcal{I}}(\boldsymbol{\xi}_t) = \sum_{\ell=1}^{L} w_{\boldsymbol{y},t-1}^{(\ell)} \log \frac{\widehat{L}_{\boldsymbol{\theta},\boldsymbol{\xi}_t}^{N}(\widetilde{\boldsymbol{y}}_t^{(\ell)})}{\widehat{Z}_{\boldsymbol{\xi}_t}^{M,N}(\widetilde{\boldsymbol{y}}_t^{(\ell)})}. \qquad (9)$$

Following a similar procedure, the gradients $\nabla_{\boldsymbol{\xi}_t} L_{\boldsymbol{\theta},\boldsymbol{\xi}_t}(\boldsymbol{y}_t)$ and $\nabla_{\boldsymbol{\xi}_t} Z_{\boldsymbol{\xi}_t}(\boldsymbol{y}_t)$ of (6) can be approximated. See details in Appendix B.

### 3.4. Consistency of the EIG Estimator

Consistency of our estimator $\widehat{\mathcal{I}}(\boldsymbol{\xi}_t)$ follows from two ingredients: (i) NPF convergence under standard conditions (small/regular jittering kernels, Lipschitz state posterior in $\boldsymbol{\theta}$, and a bounded positive likelihood), and (ii) NMC consistency provided the EIG integrand, $\log(L/Z)$, is Lipschitz and square-integrable. Formal statements and practical interpretation of these assumptions are given in Appendix C.

**Theorem 3.1.** *Let $\widehat{\mathcal{I}}(\boldsymbol{\xi}_t)$ denote the NMC estimator of the EIG (9), constructed using $M$ parameter particles, $N$ state particles per parameter, and $L$ pseudo-observations. Under Assumptions C.2–C.6 (in Appendix C), for any $t$ and $\boldsymbol{\xi}_t$*

$$\widehat{\mathcal{I}}(\boldsymbol{\xi}_t) \xrightarrow[L,M,N \to \infty]{a.s.} \mathcal{I}(\boldsymbol{\xi}_t).$$

The proof combines NMC results (Rainforth et al., 2018), ensuring consistency of nested expectations under Assumption C.6, with NPF convergence guarantees (Crisan & Míguez, 2018), which establish almost-sure convergence of empirical state–parameter measures under Assumptions C.2–C.5. Together, these results imply that the Monte Carlo averages in our estimator converge almost surely to the true EIG. Full details are given in Appendix C.

### 3.5. Overall Algorithm

BAD-PODS (Bayesian Adaptive Design for Partially Observable Dynamical Systems), selects designs sequentially at each time step $t$. Given the current history $h_{t-1}$, we optimise $\boldsymbol{\xi}_t$ via SGA (Section 3.2), using the gradient estimator of the EIG computed with NMC samples from $\Gamma_{\boldsymbol{\xi}_t}$ (Section 3.3). After $K$ optimisation steps, with step sizes $\eta_k$, we obtain the optimised design $\boldsymbol{\xi}_t^{\star}$ and the real observation $\boldsymbol{y}_t$ is collected. The posterior is then updated via an NPF (Section 2.4): parameter particles are jittered, states are propagated through $f$, and weights are adjusted using $g$, with resampling to maintain diversity. This recursive construction avoids reprocessing past data, with a per-step cost of $\mathcal{O}((KL+1)MN)$ (and $\mathcal{O}(KLMNt)$ over $t$ steps),

---

[1] Note that the dotted and double-dotted notation is used only to distinguish distinct sample sets and does not denote derivatives.

ensuring linear scaling in time while supporting gradient-based design optimisation.

The algorithm takes as input the prior distributions, particle numbers $(M, N)$, and optimisation hyperparameters $(K, \eta_k)$, and produces as output the optimised design sequence $\{\boldsymbol{\xi}_t^{\star}\}_{t=1:T}$ together with posterior approximations of states and parameters. See Appendix D for full implementation details.

## 4. Related Work

A central challenge in BED is the cost and complexity of estimating the EIG (or its gradient) (Rainforth et al., 2024; Huan et al., 2024). For models with tractable likelihoods $p(\boldsymbol{y} \,|\, \boldsymbol{\theta}, \boldsymbol{\xi})$, the NMC estimator has been studied in depth (Rainforth et al., 2018), with subsequent work proposing variational formulations and bounds to improve convergence (Foster et al., 2019; 2020; 2021). These approaches also enable differentiable objectives for gradient-based design in continuous spaces. However, they rely on explicit pointwise likelihood evaluation and do not extend directly to partially observable dynamical systems, where likelihoods require marginalisation over latent states.

When likelihoods are implicit (intractable pointwise but possible to simulate from), likelihood-free design methods replace direct evaluation by surrogates or simulation-based objectives. Examples include variational mutual-information bounds and amortised estimators (Kleinegesse & Gutmann, 2020; 2021; Ivanova et al., 2021), density-ratio or classifier-based estimates of information gain (Kleinegesse & Gutmann, 2019), and approximate Bayesian computation (ABC)-style utilities (Dehideniya et al., 2018; Drovandi & Pettitt, 2013; Hainy et al., 2016; Price et al., 2016). While effective in simulator-based settings, many likelihood-free strategies target *static* BED, where the design sequence is optimised before deployment. Amortised variants (e.g., Ivanova et al., 2021) enable sequential decisions by training a policy or surrogate offline over simulated histories and reusing it at deployment, although coverage can become challenging in long-horizon partially observed systems. In contrast, our method optimises each new design online from the current posterior induced by the data collected so far. Thus, amortised methods offer fast deployment after training, with performance depending on the coverage of the offline training distribution, whereas our method avoids offline training and optimises designs online at the cost of higher per-step computation.

Design for dynamical systems has also been studied under full state observability. Iqbal et al. (2024a;b) consider sequential BED settings in which the state is fully observed, making the likelihood tractable. Both papers leverage particle methods for joint state–parameter inference: Iqbal et al.

(2024b) use a reversed ("inside-out") SMC$^2$, and Iqbal et al. (2024a) adopt NPF's jittering to rejuvenate parameter particles. However, these recent advances in BED for dynamical systems assume full state observability and do not address the latent-state marginalisations required in partially observable SSMs.

# 5. Experiments

We evaluate our method on two partially observable dynamical systems. Since no existing methods target this setting, we compare against three standard BED baselines (Foster, 2021): (i) *random* designs sampled uniformly from $\Omega$, (ii) an *oracle* version of BAD-PODS, which (instead of gradient-based optimisation) evaluates the estimated EIG on a dense discretisation of $\Omega$ and selects the best grid point at each step (providing an approximate upper bound); and (iii) *static* BED, a non-adaptive version of our method where the full sequence $\boldsymbol{\xi}_{1:T}$ is optimised offline. Unlike our sequential approach, the static baseline cannot adapt as data arrive.

Performance is measured by the *total EIG* (TEIG), the accumulated information gain across time. We also report *relative improvements*,

$$\Delta\text{TEIG}^{(\text{baseline})} = \sum_{s=1}^{t} \left( \widehat{\mathcal{I}}(\boldsymbol{\xi}_s^{\text{BAD-PODS}}) - \widehat{\mathcal{I}}(\boldsymbol{\xi}_s^{\text{baseline}}) \right),$$

against each baseline to highlight adaptive gains.

We further assess inference accuracy using the root normalised mean squared error (RNMSE). For a quantity of interest $\boldsymbol{z}$ with ground truth $\boldsymbol{z}^\star$ and posterior mean estimate $\widehat{\boldsymbol{z}}$, we define

$$\text{RNMSE}(\boldsymbol{z}) = \sqrt{\frac{\|\widehat{\boldsymbol{z}} - \boldsymbol{z}^\star\|^2}{\|\boldsymbol{z}^\star\|^2}}.$$

We report this metric for both the static parameters, $\text{RNMSE}_\theta$, and the latent states, $\text{RNMSE}_x$, providing complementary diagnostics of parameter learning and state tracking. Results are averaged over 50 seeds with bootstrap 95% confidence intervals.

Python code to reproduce all experiments is released at https://github.com/sarapv/badpods. Details of the computing infrastructure are provided in Appendix H.

## 5.1. Two-group SIR Model

The susceptible-infectious-recovered (SIR) model is a standard testbed in BED, but prior work often assumes fully observed states (Ivanova et al., 2021) or static design (Kleinegesse & Gutmann, 2019). Here we use a *partially observable*, stochastic two-group SIR model with *online* sequential design. The latent state tracks the susceptible/infectious

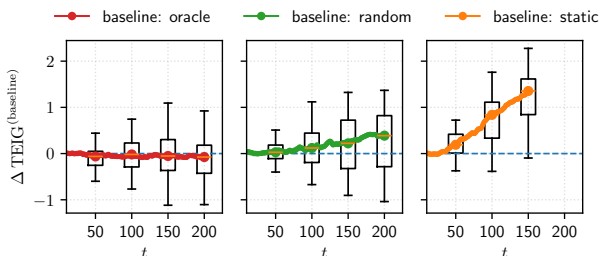

*Figure 1.* $\Delta$TEIG for the two-group SIR. Boxplots (median and interquartiles) compare BAD-PODS with the oracle (left, red), random designs (middle, green), and static BED (right, orange). Higher values indicate better performance for BAD-PODS.

counts in each group $\mathbf{X}^{(g)}(\tau) = \left(S^{(g)}(\tau), I^{(g)}(\tau)\right)^\top$, and the unknown parameters are the infection and recovery rates $\boldsymbol{\theta} = \{(\beta^{(g)}, \gamma^{(g)})\}_{g=1}^2$. At each time step $t$, the design $\boldsymbol{\xi}_t = (\xi_t^{(1)}, \xi_t^{(2)})$ allocates a fixed sampling effort across the two groups (simplex constraint), producing noisy incidence observations. The design, therefore, modulates the observation process, while the epidemic dynamics remain unchanged. Full model details are given in Appendix E.

**Experimental setup.** We set the horizon to $T = 200$ and simulate ground-truth trajectories from the two-group SIR model with group sizes $N^{(1)} = N^{(2)} = 200$ and initial infections $I_0^{(1)} = I_0^{(2)} = 5$. The cross-group mixing matrix $M$, detection scales $\rho^{(g)}$, sampling budget $\kappa$, and the parameters of the second group $(\beta^{(2)}, \gamma^{(2)})$ are fixed and known. Inference targets only the first-group parameters $(\beta^{(1)}, \gamma^{(1)})$. At each time step, the design $\boldsymbol{\xi}_t$ allocates observation effort between $y_t^{(1)}$ and $y_t^{(2)}$. For more details see Appendix E.

**Experimental results.** Table 1 reports TEIG for all four methods. BAD-PODS consistently performs close to the oracle solution, and outperforms both static and random baselines, with gaps widening over time. Results for the static method are truncated as optimisation becomes infeasible at longer horizons.

Figure 1 plots $\Delta$TEIG, confirming that gains accumulate over time. The improvements of BAD-PODS relative to the static and random baselines grow steadily with time, and are particularly higher with respect to the static baseline. Figure 2 further explains this behaviour: while random designs are highly variable, both static and sequential approaches converge to structured allocations, with BAD-PODS consistently allocating more effort to group 1 (the group with unknown parameters). Additional results based on the RNMSE for the latent states and parameters are reported in Appendix E.3, and are consistent with the results observed above.

*Table 1.* TEIG across time for the two-group SIR model (top), moving-source model (middle), and ecological growth model (bottom). Means over 50 seeds with 95% bias-corrected and accelerated (BCa) CIs. Static results are truncated at longer horizons due to computational cost.

| | | **Two-group SIR model** | | |
| --- | --- | --- | --- | --- |
| $t$ | Oracle | BAD-PODS | Random | Static |
| 50 | 1.450 | **1.390** | 1.364 | 1.167 |
| | [1.376,1.520] | [1.324,1.460] | [1.287,1.445] | [1.117,1.220] |
| 100 | 3.330 | **3.283** | 3.152 | 2.529 |
| | [3.215,3.442] | [3.173,3.380] | [3.040,3.268] | [2.444,2.621] |
| 150 | 4.848 | **4.748** | 4.518 | 3.547 |
| | [4.747,4.959] | [4.609,4.871] | [4.385,4.627] | [3.454,3.658] |
| 200 | 6.043 | **5.918** | 5.621 | – |
| | [5.939,6.168] | [5.765,6.055] | [5.466,5.793] | |

| | | **Moving source model** | | |
| --- | --- | --- | --- | --- |
| $t$ | Oracle | BAD-PODS | Random | Static |
| 10 | 0.304 | **0.297** | 0.242 | 0.270 |
| | [0.296,0.312] | [0.289,0.308] | [0.222,0.263] | [0.258,0.284] |
| 20 | 0.698 | **0.688** | 0.582 | 0.570 |
| | [0.672,0.731] | [0.661,0.726] | [0.531,0.649] | [0.541,0.604] |
| 30 | 1.158 | **1.130** | 0.940 | – |
| | [1.103,1.223] | [1.075,1.197] | [0.855,1.044] | |
| 40 | 1.515 | **1.465** | 1.217 | – |
| | [1.451,1.591] | [1.400,1.556] | [1.136,1.344] | |
| 50 | 1.805 | **1.733** | 1.435 | – |
| | [1.732,1.888] | [1.658,1.840] | [1.354,1.555] | |

| | | **Ecological growth model** | | |
| --- | --- | --- | --- | --- |
| $t$ | Oracle | BAD-PODS | Random | Static |
| 5 | 0.935 | **0.902** | 0.725 | 0.897 |
| | [0.896, 0.970] | [0.854, 0.946] | [0.657, 0.790] | [0.866, 0.946] |
| 10 | 1.618 | **1.579** | 1.337 | 1.571 |
| | [1.554, 1.690] | [1.499, 1.650] | [1.245, 1.429] | [1.507, 1.627] |
| 15 | 2.277 | **2.215** | 1.972 | – |
| | [2.175, 2.379] | [2.113, 2.312] | [1.844, 2.109] | |
| 20 | 2.621 | **2.545** | 2.344 | – |
| | [2.492, 2.741] | [2.422, 2.673] | [2.211, 2.499] | |

**Runtime and memory.** For the SIR experiment with $T = 150$, average wall-clock runtimes are 2.09 minutes for the random baseline, 31.05 minutes for the static baseline, and 12.39 minutes for BAD-PODS, yielding an approximate $2.5\times$ speed-up over the static approach. Memory is an equally important bottleneck: at $T = 150$, the static baseline uses approximately 25 GB GPU and 25 GB CPU memory, whereas BAD-PODS uses approximately 10 GB GPU and 1 GB CPU. This gap grows with $T$, since BAD-PODS only needs to store the current $M \times N$ particle set, while the static baseline must store and optimise over the full design tensor and latent trajectories of size $M \times N \times T$, making long horizons increasingly expensive.

### 5.2. Moving Source Location

Source localisation is a standard BED testbed (Ivanova et al., 2021; Foster et al., 2021), typically studied with a static source and sensor positions as designs. Here, we use a more challenging *moving* source, with designs given by the *orientations* of fixed sensors. This setup is closely related to sensor-control/target-tracking, where sensor configurations are adapted online to improve *state estimation* (Koch, 2016).

A latent source state $\boldsymbol{x}_t = (p_{x,t}, p_{y,t}, \phi_t)^\top$ evolves accord-

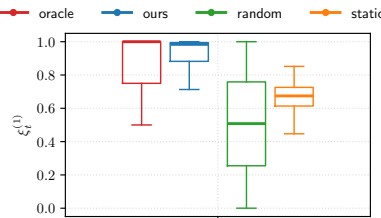

*Figure 2.* Distribution of design component $\xi_t^{(1)}$ across seeds and time (boxplots: median and interquartiles). BAD-PODS allocates more effort to group 1, which contains the unknown parameters.

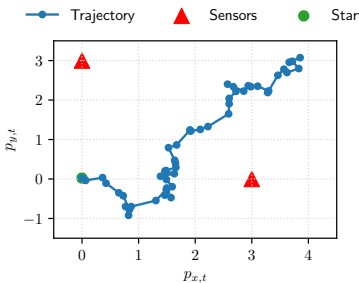

*Figure 3.* Example trajectory of the moving source for $T = 50$ (blue line) together with fixed sensor locations (red triangles).

ing to a motion model with unknown parameters $\boldsymbol{\theta} = (v_x, v_y, v_\phi)$. At each time $t$, the design $\boldsymbol{\xi}_t \in [-\pi, \pi)^J$ selects the orientation of $J$ sensors, and each sensor returns a noisy intensity whose mean depends on distance and a directional gain term that peaks when the sensor points toward the source. Hence, $\boldsymbol{\xi}_t$ affects only the observation model. Full dynamics and observation model details are in Appendix F.

**Experimental setup.** We simulate a moving source over a horizon of $T = 50$ steps starting at position $\boldsymbol{x}_0 = (0, 0, 0)$. The source parameters to be inferred are the horizontal and vertical velocities, $\boldsymbol{\theta} = (v_x, v_y)$, while the angular velocity $v_\phi$ and all other observation parameters are treated as known. We deploy $J = 2$ fixed sensors located at $\boldsymbol{s}_1 = (0, 3)$ and $\boldsymbol{s}_2 = (3, 0)$, each with orientation design variables $\boldsymbol{\xi}_t = (\xi_{t,1}, \xi_{t,2}) \in [-\pi, \pi)^2$. Figure 3 illustrates the setup and one example trajectory. Additional implementation details and parameter values are given in Appendix F.

**Experimental results.** Table 1 reports the TEIG for our sequential method compared against the oracle, the random, and the static baselines. Once again, our approach achieves the highest TEIG across all horizons, performing very close to the oracle and confirming the benefits of adapting designs online in partially observable settings. Figure 4 illustrates these results more clearly by showing boxplots of TEIG differences relative to each baseline. The performance of BAD-PODS is very close to the oracle (almost flat), and the improvements w.r.t. the other baselines accumulate steadily over time as information gain compounds. As in the SIR

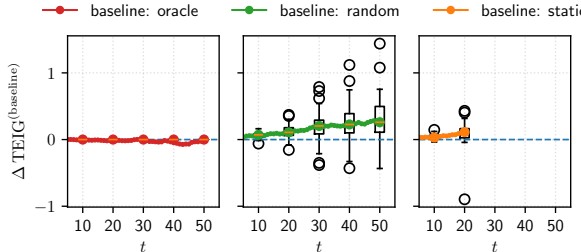

*Figure 4.* ΔTEIG for the moving source location. Boxplots (median and interquartiles) compare BAD-PODS with oracle (left, dark red), random (middle, green) and static baselines (right, orange). Higher values indicate better performance for BAD-PODS.

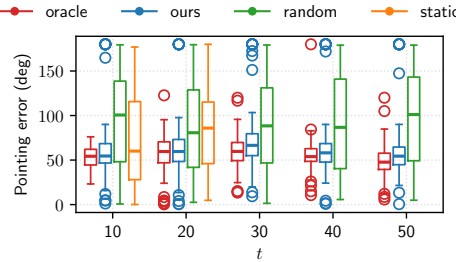

*Figure 5.* Pointing error (in degrees) of sensor orientations relative to the target at selected time steps. Boxplots (median and interquartiles) show the distribution across seeds and sensors for the oracle, our method, random design, and static BED approach. Lower values correspond to higher orientation accuracy.

experiment, results for the static method are truncated at larger horizons due to computational infeasibility of offline optimisation in high-dimensional design spaces.

Finally, we assess the quality of the selected sensor orientations in terms of their pointing error, i.e., the angular deviation between a sensor's chosen orientation and the true bearing to the source. Figure 5 shows boxplots of pointing error distributions at selected time steps across time, seeds, and sensors. Both random and static baselines exhibit wide variability, often pointing far away from the target. In contrast, our sequential method consistently achieves much smaller pointing errors, similar to those obtained with the oracle, demonstrating its ability to align sensors toward the target more frequently and with greater accuracy. However, perfect accuracy is not achieved since outliers are expected in this experiment. This is due to the signal-to-noise ratio, which sometimes is intrinsically low (e.g., the source is far and observations are dominated by background/noise), so pointing becomes weakly identifiable. Additional results based on the RNMSE for the latent states and parameters are reported in Appendix F.3.

**Runtime and memory.** For the moving-source experiment with $T = 20$, average wall-clock runtimes are 2 minutes for the random baseline, 628 minutes for the static baseline, and 525 minutes for BAD-PODS. Thus, BAD-PODS remains faster than the static baseline, but the speedup is smaller than in the other experiments since inference is more challenging and requires larger particle budgets. Memory usage is nevertheless reduced: the static baseline uses approximately 15 GB GPU and 5 GB CPU memory, whereas BAD-PODS uses approximately 13 GB GPU and 1 GB CPU. This illustrates the main computational trade-off of BAD-PODS: it avoids storing and optimising over the full design horizon, but its per-step runtime can still be substantial when accurate filtering and low-variance EIG estimates require large $M$ and $N$.

### 5.3. Ecological Growth Model

We next consider a growth/harvest SSM, commonly used in ecology to represent the evolution of a population under harvesting pressure (Zhou et al., 2009). Unlike the previous examples, where the design enters only through the observation model, here, the design affects *both* the transition dynamics and the observation process, yielding a more general testbed for sequential BED.

The latent state $x_t \in \mathbb{R}_+$ denotes the (scaled) population size at time $t$, and the design $\xi_t \in [0, 1]$ controls the harvesting intensity/effort (e.g., fishing effort or trapping rate). Observations provide a noisy, partially saturated measurement of the harvested amount. Full model details are in Appendix G.

**Experimental setup.** We simulate the growth/harvest model over a horizon of $T = 20$. The initial latent state is set to $x_0 = 0.4k$, where $r = 0.5$ and $k = 300$. The parameters to be inferred are the intrinsic growth rate and the carrying capacity $\boldsymbol{\theta} = (r, k)$, and all other observation parameters are treated as known. Additional implementation details and parameter values are provided in Appendix G.

**Experimental results.** Table 1 reports TEIG for all four methods. As in the other models, BAD-PODS consistently performs close to the oracle, and outperforms both static and random baselines. Results for the static method are truncated as optimisation becomes infeasible at longer horizons.

Figure 6 shows relative improvements in accumulated information, ΔTEIG, against each baseline. BAD-PODS achieves positive gains over random designs across the horizon, with the improvement tending to plateau toward $T = 20$. This is expected: once the posterior concentrates, the marginal information gain per additional step diminishes, and the best designs become less distinguishable. BAD-PODS shows a smaller gap or improvement w.r.t. the static baseline, since a one-shot static optimisation can already recover a reasonable policy when $T$ is short.

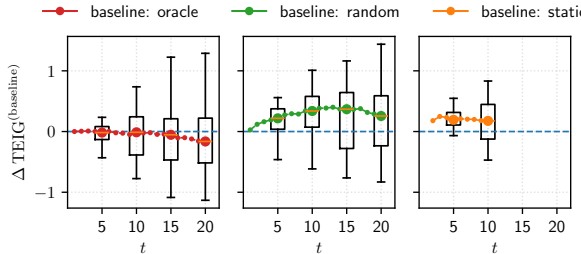

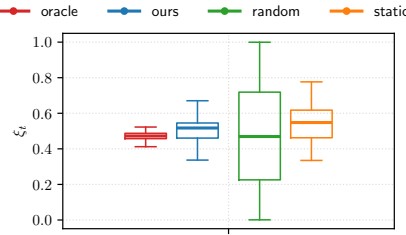

*Figure 6.* $\Delta$TEIG for the ecological growth model. Boxplots (median and interquartiles) compare BAD-PODS with oracle (left, red), random (middle, green), and static baselines (right, orange). Higher values indicate better performance for BAD-PODS.

*Figure 7.* Distribution of design $\xi_t$ across seeds and time (boxplots: median and interquartiles) for the ecological growth model. Best designs favour intermediate harvesting efforts, balancing weak signals at small $\xi_t$ against population depletion and saturated, low-sensitivity observations at large $\xi_t$.

Figure 7 reports the distribution of selected designs $\xi_t$ (aggregated over seeds and time). Both the oracle (grid search) and BAD-PODS concentrate around an intermediate harvesting effort ($\xi_t \approx 0.5$), while random designs spread across the full interval. This "sweet spot" arises from a trade-off: small $\xi_t$ produces weak harvest signals (low signal-to-noise, hence low information), whereas large $\xi_t$ quickly depresses the population state $x_t$ and drives the observation into a saturated/low-sensitivity regime. Although the selected designs are approximately stationary, the experiment still requires sequential latent-state inference, since the unobserved population trajectories evolve and can diverge across realisations. Additional results based on the RNMSE for the latent states and parameters are reported in Appendix G.3.

**Runtime and memory.** For the ecological growth experiment with $T = 10$, average wall-clock runtimes are 32 seconds for the random baseline, 311 minutes for the static baseline, and 25 minutes for BAD-PODS. Although BAD-PODS performs online optimisation at each step, it is substantially faster than the static baseline in this experiment, which requires a single optimisation over the full design horizon. Memory usage shows a similar trend: the static baseline uses approximately 25 GB GPU and 2 GB CPU memory, whereas BAD-PODS uses approximately 13 GB GPU and 1 GB CPU. This reflects the fact that BAD-PODS only maintains the current particle approximation, while the static baseline stores and optimises over the full design and latent trajectory tensors.

## 6. Conclusions

We introduced an online sequential Bayesian experimental design (BED) method for partially observable dynamical systems, modeled as state-space models (SSMs). Our approach combines two key ingredients: (i) new estimators of the expected information gain and its gradient that explicitly account for latent states, and (ii) an NPF-based construction that reuses state–parameter particles to evaluate these quantities online. This yields an algorithm whose cost

grows only linearly with time, avoids reprocessing past data, and remains asymptotically consistent under mild assumptions. The method supports continuous design spaces and stochastic-gradient optimisation, making it broadly applicable. Through experiments on a multi-group epidemiological model, a dynamical source–tracking problem and an ecological growth model, we demonstrated that online adaptation improves information efficiency over static BED or random design baselines.

**Limitations and future research.** The method inherits some limitations of nested Monte Carlo and particle filtering. Although BAD-PODS avoids replaying the full observation history and has constant computational cost at each time step, its per-step cost can still be substantial, since estimating the expected information gain (EIG) and its gradient requires Monte Carlo sampling over both latent states and parameters. In practice, performance depends on choosing sufficiently large particle budgets $(M, N)$, an adequate optimisation budget $K$, as well as a suitable jittering variance, which trades off parameter-space exploration against stability of the conditional filters. This also makes scalability to high-dimensional state or parameter spaces challenging, since particle methods may require rapidly increasing $M$ and $N$ to maintain accuracy. Finally, as discussed in Appendix C, the consistency result holds under regularity assumptions and in the limit $M, N, L \to \infty$. Near-degenerate likelihoods, repeated outliers, or severe model mismatch would violate these assumptions, increasing estimator variance or causing numerical failures. These limitations suggest several directions for future work. The proposed EIG and gradient estimators could be used as building blocks for other BED strategies, including amortised or hybrid methods tailored to short-to-moderate horizons, where fast deployment after training is especially valuable. Another open direction is long-horizon online BED: extending the framework to settings where the model, constraints, or downstream objective change over time, as may be expected under long horizons, would require additional forms of adaptivity in the model, inference scheme, or target utility.

## Acknowledgements

This work was supported by the Research Council of Finland Flagship programme: Finnish Center for Artificial Intelligence FCAI. SPV also acknowledges funding from the Helsinki Institute for Information Technology (HIIT). SI and SS acknowledge funding from the Research Council of Finland.

## Impact Statement

This paper contributes algorithms and theory for Bayesian experimental design. The methods are broadly applicable and may reduce the number of experiments needed in scientific and engineering workflows. Any societal consequences depend on the application, and we do not claim specific impacts beyond the methodological scope of this work.

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

## A. Derivation of the EIG Gradient

We derive a gradient of the EIG with respect to (w.r.t.) the design $\boldsymbol{\xi}_t$. The design affects both the transition and observation models, $f(\boldsymbol{x}_t \,|\, \boldsymbol{x}_{t-1}, \boldsymbol{\theta}, \boldsymbol{\xi}_t)$ and $g(\boldsymbol{y}_t \,|\, \boldsymbol{x}_t, \boldsymbol{\theta}, \boldsymbol{\xi}_t)$. For clarity, we introduce the joint distribution,

$$\Gamma(\boldsymbol{y}_t, \boldsymbol{x}_{0:t}, \boldsymbol{\theta} \,|\, \boldsymbol{\xi}_t, h_{t-1}) = p(\boldsymbol{\theta} \,|\, h_{t-1})\, p(\boldsymbol{x}_{0:t-1} \,|\, \boldsymbol{\theta}, h_{t-1})\, f(\boldsymbol{x}_t \,|\, \boldsymbol{x}_{t-1}, \boldsymbol{\theta}, \boldsymbol{\xi}_t)\, g(\boldsymbol{y}_t \,|\, \boldsymbol{x}_t, \boldsymbol{\theta}, \boldsymbol{\xi}_t).$$

Using this notation, the EIG in (5) (Section 3.1) can be compactly expressed as

$$\mathcal{I}(\boldsymbol{\xi}_t) = \mathbb{E}_{\Gamma(\cdot|\boldsymbol{\xi}_t, h_{t-1})}\left[ \log \frac{L_{\boldsymbol{\theta}, \boldsymbol{\xi}_t}(\boldsymbol{y}_t)}{Z_{\boldsymbol{\xi}_t}(\boldsymbol{y}_t)} \right], \qquad L_{\boldsymbol{\theta}, \boldsymbol{\xi}_t}(\boldsymbol{y}_t) = p(\boldsymbol{y}_t \,|\, \boldsymbol{\theta}, \boldsymbol{\xi}_t), \;\; Z_{\boldsymbol{\xi}_t}(\boldsymbol{y}_t) = p(\boldsymbol{y}_t \,|\, \boldsymbol{\xi}_t).$$

To compute its gradient, we write the expectation explicitly in integral form:

$$\nabla_{\boldsymbol{\xi}_t} \mathcal{I}(\boldsymbol{\xi}_t) = \nabla_{\boldsymbol{\xi}_t} \int\!\!\int\!\!\int p(\boldsymbol{\theta} \,|\, h_{t-1})\, p(\boldsymbol{x}_{0:t-1} \,|\, \boldsymbol{\theta}, h_{t-1})\, f(\boldsymbol{x}_t \,|\, \boldsymbol{x}_{t-1}, \boldsymbol{\theta}, \boldsymbol{\xi}_t)\, g(\boldsymbol{y}_t \,|\, \boldsymbol{x}_t, \boldsymbol{\theta}, \boldsymbol{\xi}_t) \times$$

$$\times \Big( \log p(\boldsymbol{y}_t \,|\, \boldsymbol{\theta}, \boldsymbol{\xi}_t) - \log p(\boldsymbol{y}_t \,|\, \boldsymbol{\xi}_t) \Big)\, \mathrm{d}\boldsymbol{y}_t \, \mathrm{d}\boldsymbol{x}_{0:t}\, \mathrm{d}\boldsymbol{\theta}.$$

Applying the product rule yields three contributions: from $f$, from $g$, and from the log–ratio term:

$$\nabla_{\boldsymbol{\xi}_t} \mathcal{I}(\boldsymbol{\xi}_t) = \int\!\!\int\!\!\int p(\boldsymbol{\theta}|h_{t-1}) p(\boldsymbol{x}_{0:t-1}|\boldsymbol{\theta}, h_{t-1}) \Big( \nabla_{\boldsymbol{\xi}_t} f(\boldsymbol{x}_t \,|\, \boldsymbol{x}_{t-1}, \boldsymbol{\theta}, \boldsymbol{\xi}_t) \Big)\, g(\boldsymbol{y}_t \,|\, \boldsymbol{x}_t, \boldsymbol{\theta}, \boldsymbol{\xi}_t) \times$$

$$\times \Big( \log p(\boldsymbol{y}_t \,|\, \boldsymbol{\theta}, \boldsymbol{\xi}_t) - \log p(\boldsymbol{y}_t \,|\, \boldsymbol{\xi}_t) \Big)\, \mathrm{d}\boldsymbol{y}_t \mathrm{d}\boldsymbol{x}_{0:t} \mathrm{d}\boldsymbol{\theta}$$

$$+ \int\!\!\int\!\!\int p(\boldsymbol{\theta}|h_{t-1}) p(\boldsymbol{x}_{0:t-1}|\boldsymbol{\theta}, h_{t-1}) f(\boldsymbol{x}_t \,|\, \boldsymbol{x}_{t-1}, \boldsymbol{\theta}, \boldsymbol{\xi}_t) \Big( \nabla_{\boldsymbol{\xi}_t} g(\boldsymbol{y}_t \,|\, \boldsymbol{x}_t, \boldsymbol{\theta}, \boldsymbol{\xi}_t) \Big) \times$$

$$\times \Big( \log p(\boldsymbol{y}_t \,|\, \boldsymbol{\theta}, \boldsymbol{\xi}_t) - \log p(\boldsymbol{y}_t \,|\, \boldsymbol{\xi}_t) \Big)\, \mathrm{d}\boldsymbol{y}_t \mathrm{d}\boldsymbol{x}_{0:t} \mathrm{d}\boldsymbol{\theta}$$

$$+ \int\!\!\int\!\!\int p(\boldsymbol{\theta}|h_{t-1}) p(\boldsymbol{x}_{0:t-1}|\boldsymbol{\theta}, h_{t-1}) f(\boldsymbol{x}_t \,|\, \boldsymbol{x}_{t-1}, \boldsymbol{\theta}, \boldsymbol{\xi}_t) g(\boldsymbol{y}_t \,|\, \boldsymbol{x}_t, \boldsymbol{\theta}, \boldsymbol{\xi}_t) \times$$

$$\times \Big( \nabla_{\boldsymbol{\xi}_t} \log p(\boldsymbol{y}_t \,|\, \boldsymbol{\theta}, \boldsymbol{\xi}_t) - \nabla_{\boldsymbol{\xi}_t} \log p(\boldsymbol{y}_t \,|\, \boldsymbol{\xi}_t) \Big)\, \mathrm{d}\boldsymbol{y}_t \mathrm{d}\boldsymbol{x}_{0:t} \mathrm{d}\boldsymbol{\theta}.$$

Using the relation $\nabla p = p \nabla \log p$, we express each derivative of $f$ and $g$ in terms of their log-gradients. This reformulation allows us to reassemble the entire expression as a single expectation over the joint distribution $\Gamma(\boldsymbol{y}_t, \boldsymbol{x}_{0:t}, \boldsymbol{\theta} \,|\, \boldsymbol{\xi}_t, h_{t-1})$:

$$\nabla_{\boldsymbol{\xi}_t} \mathcal{I}(\boldsymbol{\xi}_t) = \int\!\!\int\!\!\int p(\boldsymbol{\theta}|h_{t-1}) p(\boldsymbol{x}_{0:t-1}|\boldsymbol{\theta}, h_{t-1}) f(\boldsymbol{x}_t \,|\, \boldsymbol{x}_{t-1}, \boldsymbol{\theta}, \boldsymbol{\xi}_t) g(\boldsymbol{y}_t \,|\, \boldsymbol{x}_t, \boldsymbol{\theta}, \boldsymbol{\xi}_t) \times$$

$$\times \nabla_{\boldsymbol{\xi}_t} \log f(\boldsymbol{x}_t \,|\, \boldsymbol{x}_{t-1}, \boldsymbol{\theta}, \boldsymbol{\xi}_t) \Big( \log p(\boldsymbol{y}_t \,|\, \boldsymbol{\theta}, \boldsymbol{\xi}_t) - \log p(\boldsymbol{y}_t \,|\, \boldsymbol{\xi}_t) \Big)\, \mathrm{d}\boldsymbol{y}_t \mathrm{d}\boldsymbol{x}_{0:t} \mathrm{d}\boldsymbol{\theta}$$

$$+ \int\!\!\int\!\!\int p(\boldsymbol{\theta}|h_{t-1}) p(\boldsymbol{x}_{0:t-1}|\boldsymbol{\theta}, h_{t-1}) f(\boldsymbol{x}_t \,|\, \boldsymbol{x}_{t-1}, \boldsymbol{\theta}, \boldsymbol{\xi}_t) g(\boldsymbol{y}_t \,|\, \boldsymbol{x}_t, \boldsymbol{\theta}, \boldsymbol{\xi}_t) \times$$

$$\times \nabla_{\boldsymbol{\xi}_t} \log g(\boldsymbol{y}_t \,|\, \boldsymbol{x}_t, \boldsymbol{\theta}, \boldsymbol{\xi}_t) \Big( \log p(\boldsymbol{y}_t \,|\, \boldsymbol{\theta}, \boldsymbol{\xi}_t) - \log p(\boldsymbol{y}_t \,|\, \boldsymbol{\xi}_t) \Big)\, \mathrm{d}\boldsymbol{y}_t \mathrm{d}\boldsymbol{x}_{0:t} \mathrm{d}\boldsymbol{\theta}$$

$$+ \int\!\!\int\!\!\int p(\boldsymbol{\theta}|h_{t-1}) p(\boldsymbol{x}_{0:t-1}|\boldsymbol{\theta}, h_{t-1}) f(\boldsymbol{x}_t \,|\, \boldsymbol{x}_{t-1}, \boldsymbol{\theta}, \boldsymbol{\xi}_t) g(\boldsymbol{y}_t \,|\, \boldsymbol{x}_t, \boldsymbol{\theta}, \boldsymbol{\xi}_t) \times$$

$$\times \Big( \nabla_{\boldsymbol{\xi}_t} \log p(\boldsymbol{y}_t \,|\, \boldsymbol{\theta}, \boldsymbol{\xi}_t) - \nabla_{\boldsymbol{\xi}_t} \log p(\boldsymbol{y}_t \,|\, \boldsymbol{\xi}_t) \Big)\, \mathrm{d}\boldsymbol{y}_t \mathrm{d}\boldsymbol{x}_{0:t} \mathrm{d}\boldsymbol{\theta}$$

$$= \mathbb{E}_{\Gamma(\cdot|\boldsymbol{\xi}_t, h_{t-1})}\Big[ \nabla_{\boldsymbol{\xi}_t} \log f(\boldsymbol{x}_t \,|\, \boldsymbol{x}_{t-1}, \boldsymbol{\theta}, \boldsymbol{\xi}_t) \Big( \log p(\boldsymbol{y}_t \,|\, \boldsymbol{\theta}, \boldsymbol{\xi}_t) - \log p(\boldsymbol{y}_t \,|\, \boldsymbol{\xi}_t) \Big)$$

$$+ \nabla_{\boldsymbol{\xi}_t} \log g(\boldsymbol{y}_t \,|\, \boldsymbol{x}_t, \boldsymbol{\theta}, \boldsymbol{\xi}_t) \Big( \log p(\boldsymbol{y}_t \,|\, \boldsymbol{\theta}, \boldsymbol{\xi}_t) - \log p(\boldsymbol{y}_t \,|\, \boldsymbol{\xi}_t) \Big)$$

$$+ \Big( \nabla_{\boldsymbol{\xi}_t} \log p(\boldsymbol{y}_t \,|\, \boldsymbol{\theta}, \boldsymbol{\xi}_t) - \nabla_{\boldsymbol{\xi}_t} \log p(\boldsymbol{y}_t \,|\, \boldsymbol{\xi}_t) \Big) \Big]$$

Finally, using $\nabla \log L = \frac{\nabla L}{L}$ and $\nabla \log Z = \frac{\nabla Z}{Z}$, we obtain the compact gradient form reported in (6) (Section 3.2):

$$\nabla_{\boldsymbol{\xi}_t} \mathcal{I}(\boldsymbol{\xi}_t) = \mathbb{E}_{\Gamma(\cdot|\boldsymbol{\xi}_t, h_{t-1})}\left[ \frac{\nabla_{\boldsymbol{\xi}_t} L_{\boldsymbol{\theta}, \boldsymbol{\xi}_t}(\boldsymbol{y}_t)}{L_{\boldsymbol{\theta}, \boldsymbol{\xi}_t}(\boldsymbol{y}_t)} - \frac{\nabla_{\boldsymbol{\xi}_t} Z_{\boldsymbol{\xi}_t}(\boldsymbol{y}_t)}{Z_{\boldsymbol{\xi}_t}(\boldsymbol{y}_t)} \right.$$

$$\left. + \log \frac{L_{\boldsymbol{\theta}, \boldsymbol{\xi}_t}(\boldsymbol{y}_t)}{Z_{\boldsymbol{\xi}_t}(\boldsymbol{y}_t)} \left( \nabla_{\boldsymbol{\xi}_t} \log f(\boldsymbol{x}_t \,|\, \boldsymbol{x}_{t-1}, \boldsymbol{\theta}, \boldsymbol{\xi}_t) + \nabla_{\boldsymbol{\xi}_t} \log g(\boldsymbol{y}_t \,|\, \boldsymbol{x}_t, \boldsymbol{\theta}, \boldsymbol{\xi}_t) \right) \right].$$

The quantities $L_{\boldsymbol{\theta}, \boldsymbol{\xi}_t}(\boldsymbol{y}_t)$ and $Z_{\boldsymbol{\xi}_t}(\boldsymbol{y}_t)$ correspond to the likelihood and evidence terms, respectively:

$$L_{\boldsymbol{\theta}, \boldsymbol{\xi}_t}(\boldsymbol{y}_t) = \mathbb{E}_{p(\boldsymbol{x}_{0:t}|\boldsymbol{\theta}, \boldsymbol{\xi}_t, h_{t-1})}\left[ g(\boldsymbol{y}_t \,|\, \boldsymbol{x}_t, \boldsymbol{\theta}, \boldsymbol{\xi}_t) \right],$$

$$Z_{\boldsymbol{\xi}_t}(\boldsymbol{y}_t) = \mathbb{E}_{p(\boldsymbol{x}_{0:t}, \boldsymbol{\theta}|\boldsymbol{\xi}_t, h_{t-1})}\left[ g(\boldsymbol{y}_t \,|\, \boldsymbol{x}_t, \boldsymbol{\theta}, \boldsymbol{\xi}_t) \right],$$

where

$$p(\boldsymbol{x}_{0:t}, \boldsymbol{\theta} \,|\, \boldsymbol{\xi}_t, h_{t-1}) = f(\boldsymbol{x}_t \,|\, \boldsymbol{x}_{t-1}, \boldsymbol{\theta}, \boldsymbol{\xi}_t)\, p(\boldsymbol{x}_{0:t-1} \,|\, \boldsymbol{\theta}, h_{t-1})\, p(\boldsymbol{\theta} \,|\, h_{t-1}),$$
$$p(\boldsymbol{x}_{0:t} \,|\, \boldsymbol{\theta}, h_{t-1}) = f(\boldsymbol{x}_t \,|\, \boldsymbol{x}_{t-1}, \boldsymbol{\theta}, \boldsymbol{\xi}_t)\, p(\boldsymbol{x}_{0:t-1} \,|\, \boldsymbol{\theta}, h_{t-1}),$$

denote the joint and conditional predictive distributions at time $t-1$.

Following the same steps used above (product rule and log-gradients), the likelihood and evidence gradients are

$$\nabla_{\boldsymbol{\xi}_t} L_{\boldsymbol{\theta}, \boldsymbol{\xi}_t}(\boldsymbol{y}_t) = \mathbb{E}_{p(\boldsymbol{x}_{0:t}|\boldsymbol{\theta}, \boldsymbol{\xi}_t, h_{t-1})}\left[ \nabla_{\boldsymbol{\xi}_t} g(\boldsymbol{y}_t \,|\, \boldsymbol{x}_t, \boldsymbol{\theta}, \boldsymbol{\xi}_t) + g(\boldsymbol{y}_t \,|\, \boldsymbol{x}_t, \boldsymbol{\theta}, \boldsymbol{\xi}_t)\, \nabla_{\boldsymbol{\xi}_t} \log f(\boldsymbol{x}_t \,|\, \boldsymbol{x}_{t-1}, \boldsymbol{\theta}, \boldsymbol{\xi}_t) \right],$$

$$\nabla_{\boldsymbol{\xi}_t} Z_{\boldsymbol{\xi}_t}(\boldsymbol{y}_t) = \mathbb{E}_{p(\boldsymbol{x}_{0:t}, \boldsymbol{\theta}|\boldsymbol{\xi}_t, h_{t-1})}\left[ \nabla_{\boldsymbol{\xi}_t} g(\boldsymbol{y}_t \,|\, \boldsymbol{x}_t, \boldsymbol{\theta}, \boldsymbol{\xi}_t) + g(\boldsymbol{y}_t \,|\, \boldsymbol{x}_t, \boldsymbol{\theta}, \boldsymbol{\xi}_t)\, \nabla_{\boldsymbol{\xi}_t} \log f(\boldsymbol{x}_t \,|\, \boldsymbol{x}_{t-1}, \boldsymbol{\theta}, \boldsymbol{\xi}_t) \right].$$

## B. Monte Carlo Approximations of Gradients

We now derive the Monte Carlo approximations used to estimate the gradient $\nabla_{\boldsymbol{\xi}_t} \mathcal{I}(\boldsymbol{\xi}_t)$ in (6) (Section 3.2). The procedure mirrors the sampling strategy introduced in Section 3.3 for the EIG itself.

**Outer expectation.** The gradient expression in (6) (Section 3.2) involves an expectation over the joint distribution $\Gamma(\boldsymbol{y}_t, \boldsymbol{x}_{0:t}, \boldsymbol{\theta} \,|\, \boldsymbol{\xi}_t, h_{t-1})$, which can be approximated with $L$ Monte Carlo samples:

$$\widehat{\nabla_{\boldsymbol{\xi}_t} \mathcal{I}}(\boldsymbol{\xi}_t) = \sum_{\ell=1}^{L} w_{\boldsymbol{y}, t}^{(\ell)}\left[ \frac{\widehat{\nabla_{\boldsymbol{\xi}_t} L}_{\boldsymbol{\theta}^{(\ell)}, \boldsymbol{\xi}_t}^{N}(\widetilde{\boldsymbol{y}}_t^{(\ell)})}{\widehat{L}_{\boldsymbol{\theta}^{(\ell)}, \boldsymbol{\xi}_t}^{N}(\widetilde{\boldsymbol{y}}_t^{(\ell)})} - \frac{\widehat{\nabla_{\boldsymbol{\xi}_t} Z}_{\boldsymbol{\xi}_t}^{M,N}(\widetilde{\boldsymbol{y}}_t^{(\ell)})}{\widehat{Z}_{\boldsymbol{\xi}_t}^{M,N}(\widetilde{\boldsymbol{y}}_t^{(\ell)})} \right.$$

$$\left. + \log \frac{\widehat{L}_{\boldsymbol{\theta}^{(\ell)}, \boldsymbol{\xi}_t}^{N}(\widetilde{\boldsymbol{y}}_t^{(\ell)})}{\widehat{Z}_{\boldsymbol{\xi}_t}^{M,N}(\widetilde{\boldsymbol{y}}_t^{(\ell)})} \left( \nabla_{\boldsymbol{\xi}_t} \log f(\widetilde{\boldsymbol{x}}_t^{(\ell)} \,|\, \boldsymbol{x}_{t-1}^{(\ell)}, \boldsymbol{\theta}_{t-1}^{(\ell)}, \boldsymbol{\xi}_t) + \nabla_{\boldsymbol{\xi}_t} \log g(\widetilde{\boldsymbol{y}}_t^{(\ell)} \,|\, \widetilde{\boldsymbol{x}}_t^{(\ell)}, \boldsymbol{\theta}_{t-1}^{(\ell)}, \boldsymbol{\xi}_t) \right) \right]. \quad (10)$$

**Sample generation.** Monte Carlo samples from $\Gamma(\cdot \,|\, \boldsymbol{\xi}_t, h_{t-1})$ form the array

$$\left\{ (\boldsymbol{\theta}_{t-1}^{(m)}, \boldsymbol{x}_{0:t-1}^{(m,n)}, \widetilde{\boldsymbol{x}}_t^{(m,n)}, \widetilde{\boldsymbol{y}}_t^{(m,n)}) \right\}, \qquad m = 1, \ldots, M, \ \ n = 1, \ldots, N,$$

with weights combining the parameter and state particle weights from time $t-1$:

$$w_{\boldsymbol{y}, t}^{(m,n)} = w_{\boldsymbol{\theta}, t-1}^{(m)}\, w_{\boldsymbol{x}, t-1}^{(m,n)}.$$

Each pair $(m, n)$ corresponds to a nested particle configuration:

$$\boldsymbol{\theta}_{t-1}^{(m)} \sim p(\boldsymbol{\theta} \,|\, h_{t-1}),$$
$$\boldsymbol{x}_{0:t-1}^{(m,n)} \sim p(\boldsymbol{x}_{0:t-1} \,|\, \boldsymbol{\theta}_{t-1}^{(m)}, h_{t-1}),$$
$$\widetilde{\boldsymbol{x}}_t^{(m,n)} \sim f(\cdot \,|\, \boldsymbol{x}_{t-1}^{(m,n)}, \boldsymbol{\theta}_{t-1}^{(m)}, \boldsymbol{\xi}_t),$$
$$\widetilde{\boldsymbol{y}}_t^{(m,n)} \sim g(\cdot \,|\, \widetilde{\boldsymbol{x}}_t^{(m,n)}, \boldsymbol{\theta}_{t-1}^{(m)}, \boldsymbol{\xi}_t).$$

For notational simplicity, we relabel the samples and weights as

$$\left\{(\boldsymbol{\theta}_{t-1}^{(\ell)}, \boldsymbol{x}_{0:t-1}^{(\ell)}, \widetilde{\boldsymbol{x}}_t^{(\ell)}, \widetilde{\boldsymbol{y}}_t^{(\ell)}, w_{\boldsymbol{y},t}^{(\ell)})\right\}, \qquad \ell = 1, \ldots, L, \quad L = M \times N, \tag{11}$$

where each index $\ell$ corresponds to a unique pair $(m, n)$. This reindexing allows for the compact summation form used in the estimator above.

**Likelihood and evidence estimates.** Each term in (10) (Appendix B) requires evaluating—and differentiating—the likelihood and evidence. Following the same nested sampling structure used for the EIG estimators (Section 3.3), we approximate these quantities using $M$ outer (parameter) samples and $N$ inner (state) samples:

$$\widehat{L}_{\boldsymbol{\theta}^{(\cdot)}, \boldsymbol{\xi}_t}^N(\boldsymbol{y}_t) = \sum_{n=1}^N w_{\boldsymbol{x},t}^{(\cdot,n)} \, g(\boldsymbol{y}_t \,|\, \ddot{\boldsymbol{x}}_t^{(\cdot,n)}, \boldsymbol{\theta}^{(\cdot)}, \boldsymbol{\xi}_t)$$

$$\text{for} \quad \ddot{\boldsymbol{x}}_t^{(\cdot,n)} \sim f(\cdot \,|\, \boldsymbol{x}_{t-1}^{(\cdot,n)}, \boldsymbol{\theta}^{(\cdot)}, \boldsymbol{\xi}_t),$$

$$\widehat{Z}_{\boldsymbol{\xi}_t}^{M,N}(\boldsymbol{y}_t) = \sum_{m=1}^M \sum_{n=1}^N w_{\boldsymbol{\theta},t}^{(m)} \, w_{\boldsymbol{x},t}^{(m,n)} \, g(\boldsymbol{y}_t \,|\, \dot{\boldsymbol{x}}_t^{(m,n)}, \dot{\boldsymbol{\theta}}_{t-1}^{(m)}, \boldsymbol{\xi}_t)$$

$$\text{for} \quad \dot{\boldsymbol{\theta}}_{t-1}^{(m)} \sim \kappa_M(\cdot \,|\, \boldsymbol{\theta}_{t-1}^{(m)}) \;\; \text{and} \;\; \dot{\boldsymbol{x}}_t^{(m,n)} \sim f(\cdot \,|\, \boldsymbol{x}_{t-1}^{(m,n)}, \dot{\boldsymbol{\theta}}^{(m)}, \boldsymbol{\xi}_t).$$

Here, the superscripts $(n)$ and $(m, n)$ emphasize that the estimators depend on the number of inner (state) and outer (parameter) particles, respectively. The weighting terms $w_{\boldsymbol{x},t}^{(m,n)}$ and $w_{\boldsymbol{\theta},t}^{(m)}$ correspond to the normalized particle weights from the posterior approximations of the NPF at time $t - 1$.

**Gradients of likelihood and evidence.** Following Appendix A, the gradients of the inner expectations can be estimated as

$$\widehat{\nabla_{\boldsymbol{\xi}_t} L}_{\boldsymbol{\theta}^{(\cdot)}, \boldsymbol{\xi}_t}^N(\boldsymbol{y}_t) = \sum_{n=1}^N w_{\boldsymbol{x},t-1}^{(\cdot,n)} \left[ \nabla_{\boldsymbol{\xi}_t} g(\boldsymbol{y}_t \,|\, \ddot{\boldsymbol{x}}_t^{(\cdot,n)}, \boldsymbol{\theta}, \boldsymbol{\xi}_t) + g(\boldsymbol{y}_t \,|\, \ddot{\boldsymbol{x}}_t^{(\cdot,n)}, \boldsymbol{\theta}, \boldsymbol{\xi}_t) \nabla_{\boldsymbol{\xi}_t} \log f(\ddot{\boldsymbol{x}}_t^{(\cdot,n)} \,|\, \boldsymbol{x}_{t-1}^{(\cdot,n)}, \boldsymbol{\theta}, \boldsymbol{\xi}_t) \right],$$

$$\widehat{\nabla_{\boldsymbol{\xi}_t} Z}_{\boldsymbol{\xi}_t}^{M,N}(\boldsymbol{y}_t) = \sum_{m=1}^M \sum_{n=1}^N w_{\boldsymbol{x},t-1}^{(m,n)} \, w_{\boldsymbol{\theta},t-1}^{(m)} \left[ \nabla_{\boldsymbol{\xi}_t} g(\boldsymbol{y}_t \,|\, \dot{\boldsymbol{x}}_t^{(m,n)}, \dot{\boldsymbol{\theta}}_{t-1}^i, \boldsymbol{\xi}_t) \right.$$

$$\left. + \, g(\boldsymbol{y}_t \,|\, \dot{\boldsymbol{x}}_t^{(m,n)}, \dot{\boldsymbol{\theta}}_{t-1}^i, \boldsymbol{\xi}_t) \nabla_{\boldsymbol{\xi}_t} \log f(\dot{\boldsymbol{x}}_t^{(m,n)} \,|\, \boldsymbol{x}_{t-1}^{(m,n)}, \dot{\boldsymbol{\theta}}_{t-1}^i, \boldsymbol{\xi}_t) \right].$$

The first expression provides a Monte Carlo estimate of the gradient of the likelihood $p(\boldsymbol{y}_t \,|\, \boldsymbol{\theta}, \boldsymbol{\xi}_t)$ under fixed parameter particles, while the second corresponds to the evidence gradient averaged over both parameter and state particles. Together, they yield estimates of the terms required in the gradient of the EIG.

## C. Proof of Consistency of the EIG Estimator

Let $D_\theta \subset \mathbb{R}^{d_\theta}$ denote the compact parameter domain. For a bounded measurable function $h : D_\theta \to \mathbb{R}$ we write $h \in B(D_\theta)$, with supremum norm $\|h\|_\infty = \sup_{\boldsymbol{\theta} \in D_\theta} |h(\boldsymbol{\theta})|$. For a distribution $\phi$ on a measurable space $\mathcal{X}$ and a test function $f : \mathcal{X} \to \mathbb{R}$, we denote $(f, \phi) = \int f(x) \, \phi(\mathrm{d}x)$. We use $\|\cdot\|_p$ for the $\ell^p$ norm on $\mathbb{R}^{d_\theta}$, and abbreviate almost sure convergence as $\xrightarrow{\text{a.s.}}$.

**Theorem C.1** (Consistency of EIG estimator). *Let $\widehat{\mathcal{I}}(\boldsymbol{\xi}_t)$ denote the nested Monte Carlo (NMC) estimator of the expected information gain defined in (9) (Section 3.3), constructed with $M$ parameter particles, $N$ state particles per parameter, and $L$ pseudo-observations. Under Assumptions C.2–C.6,*

$$\widehat{\mathcal{I}}(\boldsymbol{\xi}_t) \xrightarrow[L,M,N\to\infty]{\text{a.s.}} \mathcal{I}(\boldsymbol{\xi}_t),$$

*for any $t$ and $\boldsymbol{\xi}_t$.*

**Assumption C.2** (Jittering kernel scaling). There exist $p \geq 1$ and $c_\kappa < \infty$ such that

$$\sup_{\boldsymbol{\theta}' \in D_\theta} \int \|\boldsymbol{\theta} - \boldsymbol{\theta}'\|_p \, \kappa_M(\mathrm{d}\boldsymbol{\theta} \,|\, \boldsymbol{\theta}') \;\leq\; \frac{c_\kappa^p}{M^{p/2}}.$$

**Assumption C.3** (Jittering kernel regularity). For any $h \in B(D_\theta)$,

$$\sup_{\theta' \in D_\theta} \int |h(\theta) - h(\theta')| \, \kappa_M(d\theta \,|\, \theta') \;\leq\; \frac{c_\kappa \|h\|_\infty}{\sqrt{M}}.$$

**Assumption C.4** (Lipschitz dependence of state posteriors on $\theta$). For each $t \geq 1$, let $\phi_{t,\theta}$ denote the posterior of $x_t$ given $(y_{1:t}, \xi_{1:t})$ and parameter $\theta \in D_\theta$. Then for every bounded measurable $f : \mathbb{R}^{d_x} \to \mathbb{R}$ there exists $b_t < \infty$ such that

$$\left| (f, \phi_{t,\theta'}) - (f, \phi_{t,\theta''}) \right| \;\leq\; b_t \, \|f\|_\infty \, \|\theta' - \theta''\|, \quad \forall \theta', \theta'' \in D_\theta.$$

**Assumption C.5** (Bounded, positive likelihood). For (almost) every $y_t$ in the observation space and each $t$,

$$0 < \inf_{\theta \in D_\theta, \, x_t} g(y_t \,|\, x_t, \theta, \xi_t) \;\leq\; \sup_{\theta \in D_\theta, \, x_t} g(y_t \,|\, x_t, \theta, \xi_t) < \infty.$$

That is, $g$ is uniformly bounded above and bounded away from zero across $\theta \in D_\theta$ (and $x_t$) for any $\xi_t$.

**Assumption C.6** (Regularity of the integrand for NMC). Let $L_{\theta, \xi_t}(y_t) = \mathbb{E}_{p(x_{0:t} | \theta, h_{t-1})}[\, g(y_y \,|\, x_t, \theta, \xi_t) \,]$ and $Z_{\xi_t}(y_t) = \mathbb{E}_{p(\theta | h_{t-1}) p(x_{0:t} | \theta, h_{t-1})}[\, g(y_t \,|\, x_t, \theta, \xi_t) \,]$. The information–gain integrand

$$f(y_t, \theta, x_{0:t}) \;=\; \log \frac{L_{\theta, \xi_t}(y_t)}{Z_{\xi_t}(y_t)}$$

is (i) Lipschitz continuous in the inner random arguments passed from the inner estimators (i.e., in the latent state path and any inner re-sampled $\theta$) and (ii) square-integrable ($f \in L^2$). Assumption C.5 implies (ii), since boundedness and positivity yield $L_{\theta, \xi_t}, Z_{\xi_t} \in (\epsilon, \infty)$ and hence $\log(L_{\theta, \xi_t}/Z_{\xi_t}) \in L^2$ and is continuous on $[\epsilon, \infty)$.

*Remark* C.7. Assumptions C.2–C.5 are standard in the analysis of NPF (Crisan & Míguez, 2018) and ensure that the empirical measures of the particle system converge almost surely to the true posterior/predictive distrubutions. Assumption C.6 is the usual NMC regularity condition (Rainforth et al., 2018) , stated here for the EIG integrand.

In brief, the jittering assumptions, Assumptions C.2–C.3, control the artificial perturbation of parameter particles, while Assumptions C.4–C.6 ensure stable filtering recursions and well-behaved likelihood ratios in the EIG estimator. Among these, Assumptions C.4 and C.5 are often the most directly tied to practical model behaviour. They are expected to hold when the observation model has sufficiently regular, positive, and non-degenerate likelihoods, and when small changes in $\theta$ do not lead to abrupt changes in the filtering distributions.

If Assumption C.5 is violated, for instance, because the observation likelihood is very small or numerically zero for a non-negligible set of samples, then both $L_{\theta, \xi_t}(y_t)$ and $Z_{\xi_t}(y_t)$ can become extremely small. This can lead to unstable log-ratios in Assumption C.6, high-variance estimates or gradients, and occasional numerical failures. Such behaviour may occur, for example, with near-deterministic observations, bounded-support or truncated likelihoods, repeated outliers, or model mismatch. Increasing the Monte Carlo sizes $M, N$ reduces variance, and numerical stabilisation can help in mild cases, but these remedies do not remove the intrinsic ill-conditioning of the log-ratio. Thus, repeated violations of the positivity/boundedness condition in Assumption C.5 may invalidate the regularity of the EIG integrand required in Assumption C.6, and hence prevent the convergence result in Theorem 3.1 from applying.

Violations of Assumption C.4, on the other hand, can make the filtering recursion highly sensitive to changes in $\theta$. This can increase approximation error and variance in the inference task, which in turn increases the variance of the EIG estimator and may slow convergence.

### C.1. Proof of Theorem C.1

**Convergence of the evidence estimator.** Theorem 3 of Crisan & Míguez (2018) shows that the joint parameter–state posterior empirical measure produced by the NPF, $\widehat{\pi}_t^{M,N} = \widehat{p}(d\theta, dx_{0:t} \,|\, h_t)$, converges to the true posterior $\pi_t = p(d\theta, dx_{0:t} \,|\, h_t)$ as $M, N \to \infty$. Adapting notation to our setting:

**Theorem C.8** (Crisan and Míguez, 2018, Thm. 3). *Let $h_T = \{y_{1:T}, \xi_{1:T}\}$ be fixed and $f \in B(D_\theta \times \mathbb{R}^{d_x})$. Under Assumptions C.2–C.5, for any $p \geq 1$ and $1 \leq t \leq T$,*

$$\left\| (f, \widehat{\pi}_t^{M,N}) - (f, \pi_t) \right\|_p \;\leq\; \frac{c_t \|f\|_\infty}{\sqrt{M}} + \frac{\bar{c}_t \|f\|_\infty}{\sqrt{N}},$$

*where the constants $\{c_t, \bar{c}_t\}_{1 \leq t \leq T}$ are finite and independent of $M, N$.*

Taking $f(\boldsymbol{\theta}, \boldsymbol{x}_{0:t}) = g(\boldsymbol{y}_t \,|\, \boldsymbol{x}_t, \boldsymbol{\theta}, \boldsymbol{\xi}_t)$, we obtain

$$\widehat{Z}^{M,N}_{\boldsymbol{\xi}_t}(\boldsymbol{y}_t) \;=\; (f, \widehat{\pi}^{M,N}_t) \; \xrightarrow[M,N\to\infty]{\text{a.s.}} \; (f, \pi_t) \;=\; Z_{\boldsymbol{\xi}_t}(\boldsymbol{y}_t), \tag{12}$$

so the particle estimator of the evidence converges almost surely to the true value. In particular, any Monte Carlo estimator that integrates against $\widehat{\pi}^{M,N}_t$ (including its gradient forms) inherits these asymptotic guarantees.

**Convergence of the likelihood estimator.**   Theorem 2 and Remark 10 of Crisan & Míguez (2018) establish convergence for the empirical parameter posterior $\widehat{\mu}^{M,N}_t = \widehat{p}(\mathrm{d}\boldsymbol{\theta} \,|\, h_t)$, and for the conditional state filters $\widehat{\phi}^N_{t,\boldsymbol{\theta}'} = \widehat{p}(\mathrm{d}\boldsymbol{x}_t \,|\, \boldsymbol{\theta}', h_t)$ computed within the NPF.

**Theorem C.9** (Crisan and Míguez, 2018, Thm. 2). *Let $h_T = \{\boldsymbol{y}_{1:T}, \boldsymbol{\xi}_{1:T}\}$ be fixed ($T < \infty$) and $h \in B(\mathbb{R}^{d_\theta})$. Under Assumptions C.2–C.5, for any $p \geq 1$ and $1 \leq t \leq T$,*

$$\left\| (h, \widehat{\mu}^{M,N}_t) - (h, \mu_t) \right\|_p \;\leq\; \frac{c_t \|h\|_\infty}{\sqrt{M}} + \frac{\bar{c}_t \|h\|_\infty}{\sqrt{N}},$$

*where $\widehat{\mu}^{M,N}_t = \widehat{p}(\mathrm{d}\boldsymbol{\theta} \,|\, h_t)$ and $\mu_t = p(\mathrm{d}\boldsymbol{\theta} \,|\, h_t)$, and the constants $\{c_t, \bar{c}_t\}_{1 \leq t \leq T}$ are finite and independent of $M, N$.*

Moreover, by Remark 10 in Crisan & Míguez (2018), the same proof yields uniform error bounds for the conditional state filters $\widehat{\phi}^N_{t,\boldsymbol{\theta}'}$ associated with each parameter particle: letting $\phi_{t,\boldsymbol{\theta}'} = p(\mathrm{d}\boldsymbol{x}_t \,|\, \boldsymbol{\theta}', h_t)$ and any $f \in B(\mathbb{R}^{d_x})$,

$$\sup_{1 \leq m \leq M} \left\| (f, \widehat{\phi}^N_{t,\boldsymbol{\theta}'}) - (f, \phi_{t,\boldsymbol{\theta}'}) \right\|_p \;\leq\; \frac{k_t \|f\|_\infty}{\sqrt{M}} + \frac{\bar{k}_t \|f\|_\infty}{\sqrt{N}},$$

for some finite constants $k_t, \bar{k}_t$ independent of $M, N$. Choosing $f(\boldsymbol{x}_t) = g(\boldsymbol{y}_t \,|\, \boldsymbol{x}_t, \boldsymbol{\theta}, \boldsymbol{\xi}_t)$, we obtain, for each $m$ parameter sample in the NPF,

$$\widehat{L}^N_{\boldsymbol{\theta}^{(m)},\boldsymbol{\xi}_t}(\boldsymbol{y}_t) \;=\; (f, \widehat{\phi}^N_{t,\boldsymbol{\theta}^{(m)}}) \; \xrightarrow[N\to\infty]{\text{a.s.}} \; (f, \phi_{t,\boldsymbol{\theta}^{(m)}}) \;=\; L_{\boldsymbol{\theta}^{(m)},\boldsymbol{\xi}_t}(\boldsymbol{y}_t). \tag{13}$$

**Convergence of the NMC estimator.**   Finally, Theorem 1 of Rainforth et al. (2018) gives consistency of NMC estimators under mild regularity. In our setting, the outer level averages over $\Gamma(\boldsymbol{y}_t, \boldsymbol{x}_{0:t}, \boldsymbol{\theta} \,|\, \boldsymbol{\xi}_t, h_{t-1})$ and the inner levels compute $L_{\boldsymbol{\theta},\boldsymbol{\xi}_t}(\boldsymbol{y}_t)$ and $Z_{\boldsymbol{\xi}_t}(\boldsymbol{y}_t)$. Up to some rewriting, the theorem is:

**Theorem C.10** (Rainforth et al., 2018, Thm. 1). *Let $f\big(y, \gamma_L(y), \gamma_Z(y)\big) = \log\big(\gamma_L(y)/\gamma_Z(y)\big)$, with $\gamma_L(y) = \mathbb{E}_{p(x)}[g(y \,|\, x)]$ and $\gamma_Z(y) = \mathbb{E}_{p(x,\theta)}[g(y \,|\, x, \theta)]$. If $f$ is Lipschitz and $f\big(y, \gamma_L(y), \gamma_Z(y)\big)$, $g(y \,|\, x)$, $g(y \,|\, x, \theta) \in L^2$, then the NMC estimator*

$$\widehat{I}^{L,M,N} = \frac{1}{L} \sum_{\ell=1}^{L} f\left( y^{(\ell)}, \frac{1}{N} \sum_{n=1}^{N} g(y^{(\ell)} \,|\, x^{(\ell,n)}), \frac{1}{MN} \sum_{m=1}^{M} \sum_{n=1}^{N} g(y^{(\ell)} \,|\, x^{(\ell,m,n)}, \theta^{(\ell,m)}) \right) \; \xrightarrow[L,M,N\to\infty]{\text{a.s.}} \; I$$

In our formulation, $g$ is the observation model $g(\boldsymbol{y}_t \,|\, \boldsymbol{x}_t, \boldsymbol{\theta}, \boldsymbol{\xi}_t)$, which uses latent states $\boldsymbol{x}_t$ for the likelihood term, and uses both $(\boldsymbol{x}_t, \boldsymbol{\theta})$ for the evidence. Assumption C.6 (regularity of the integrand) ensures Lipschitz continuity and square-integrability. Combining (12)–(13) (Appendix C) with Theorem C.10, we conclude that, for any $t$ and $\boldsymbol{\xi}_t$,

$$\widehat{\mathcal{I}}(\boldsymbol{\xi}_t) \; \xrightarrow[L,M,N\to\infty]{\text{a.s.}} \; \mathcal{I}(\boldsymbol{\xi}_t).$$

$\square$

# D. Algorithm Details

This appendix provides complete details of BAD-PODS and the baselines we compare with. Algorithm 1 presents the full BAD-PODS pipeline, including stochastic gradient–based design optimisation and sequential inference via an NPF. At each time $t$, BAD-PODS optimises the design by maximising the EIG using stochastic gradient ascent (SGA), executes the experiment at that design, and updates the joint parameter–state posterior.

For comparison, the baselines share the same inference update as BAD-PODS and differ only in design selection:

- **Random design:** identical to Algorithm 1 but without the optimisation loop. Each $\boldsymbol{\xi}_t^\star$ is sampled randomly.

- **Oracle (grid search):** identical to Algorithm 1 but replaces the stochastic-gradient loop with an exhaustive search over a fixed grid of candidate designs. At each time step, we evaluate the estimated EIG for all grid points and set $\boldsymbol{\xi}_t^\star$ to the maximiser. This provides an expensive but informative upper-bound reference in low-dimensional design spaces.

- **Static BED:** performs a single offline optimisation of the whole sequence $\{\boldsymbol{\xi}_1, \ldots, \boldsymbol{\xi}_T\}$ and then runs the same online inference using those fixed designs.

### D.1. Practical tuning and diagnostics

The performance of BAD-PODS depends on the quality of the nested particle approximation, and in practice the jittering variance and the particle budgets $(M, N)$ are the most important tuning choices. In the nested particle filter (NPF), jittering controls both exploration of the parameter space and stability of the conditional state filters. If the jittering variance is too small, parameter exploration can be slow and the posterior approximation may collapse onto a small number of parameter particles. If it is too large, the perturbed parameters may become inconsistent with the associated conditional state-particle approximations, increasing filtering error and potentially leading to weight degeneracy. Since the appropriate scale depends on the model, we treat the jittering variance as a model-dependent tuning parameter.

Increasing the number of parameter particles $M$ and state particles $N$ improves the accuracy of the posterior, EIG, and gradient estimates, but increases runtime and memory. Insufficient particle budgets typically appear through weight degeneracy. We therefore monitor the effective sample size (ESS) at both levels of the nested particle approximation: the parameter-particle ESS and the conditional state-particle ESS. Persistently low ESS suggests insufficient $M$ and/or $N$, or poorly tuned jittering. As an additional offline diagnostic, one can rerun a subset of experiments with increasing $M$ and $N$ and check whether the posterior and EIG approximations stabilise.

## E. Two-group SIR Model

### E.1. Model Description

While the formulation applies to an arbitrary number of subpopulations, we focus here on the two-group case. Let the stacked state be $\mathbf{X}(\tau) = \left(S^{(1)}(\tau),\, I^{(1)}(\tau),\, S^{(2)}(\tau),\, I^{(2)}(\tau)\right)^\top \in \mathbb{R}^4$, where $S$ and $I$ stand for *susceptible* and *infectious*, respectively. Populations $N_g$ are constant, so the number of *recovered* is $R^{(g)}(\tau) = N^{(g)} - S^{(g)}(\tau) - I^{(g)}(\tau)$. Parameters are $\boldsymbol{\theta} = \{(\beta^{(g)}, \gamma^{(g)})\}_{g=1}^2$ are infection and recovery rate constants for each group $g$. Cross-group mixing is specified by a fixed $\mathbf{M} \in \mathbb{R}^{2 \times 2}$ (e.g., nonnegative, rows summing to one).

Given the current state $\mathbf{X}$, the corresponding total (population-level) transition rates are

$$\lambda^{(g)}(\mathbf{X}) \;=\; \beta^{(g)}\, S^{(g)} \sum_{h=1}^{2} \mathbf{M}_{gh}\, \frac{I^{(h)}}{N^{(h)}} \qquad \text{and} \qquad r^{(g)}(\mathbf{X}) \;=\; \gamma^{(g)}\, I^{(g)}. \tag{14}$$

where $\lambda^{(g)}(\mathbf{X})$ is the total rate of new infections and $r^{(g)}(\mathbf{X})$ the total recovery rate in group $g$.

**Stochastic differential equation (SDE).**  Let $\mathbf{W}(\tau) \in \mathbb{R}^4$ be a standard Wiener process. The Itô SDE is

$$\mathrm{d}\mathbf{X}(\tau) \;=\; f\big(\mathbf{X}(\tau)\big)\, \mathrm{d}\tau \;+\; G\big(\mathbf{X}(\tau)\big)\, \mathrm{d}\mathbf{W}(\tau), \tag{15}$$

with drift $f(\mathbf{X}) = S\, a(\mathbf{X})$ and diffusion factor $G(\mathbf{X}) = S \operatorname{diag}\big(\sqrt{a(\mathbf{X})}\big)$, where

$$S \;=\; \begin{pmatrix} -1 & 0 & 0 & 0 \\ 1 & -1 & 0 & 0 \\ 0 & 0 & -1 & 0 \\ 0 & 0 & 1 & -1 \end{pmatrix} \qquad \text{and} \qquad a(\mathbf{X}) \;=\; \left(\lambda^{(1)}(\mathbf{X}),\, r^{(1)}(\mathbf{X}),\, \lambda^{(2)}(\mathbf{X}),\, r^{(2)}(\mathbf{X})\right)^\top.$$

**Euler–Maruyama discretization.**  For step $\Delta\tau > 0$,

$$\mathbf{X}_{t+1} \;=\; \mathbf{X}_t \;+\; S\, a(\mathbf{X}_t)\, \Delta\tau \;+\; S \operatorname{diag}\big(\sqrt{a(\mathbf{X}_t)}\big)\, \Delta\mathbf{W}_t, \qquad \Delta\mathbf{W}_t \sim \mathcal{N}\big(\mathbf{0},\, \Delta\tau\, \mathbf{I}_4\big). \tag{16}$$

To preserve feasibility, project to the per-group simplex of counts:

$$S_{t+1}^{(g)} \leftarrow \min\{N^{(g)}, \max\{0, S_{t+1}^{(g)}\}\},$$
$$I_{t+1}^{(g)} \leftarrow \min\{N^{(g)} - S_{t+1}^{(g)}, \max\{0, I_{t+1}^{(g)}\}\},$$

for $g \in \{1, 2\}$, and optionally set $R_{t+1}^{(g)} = N^{(g)} - S_{t+1}^{(g)} - I_{t+1}^{(g)}$.

**Observation model and design.** At time $t$, choose $\boldsymbol{\xi}_t = \big(\xi_t^{(1)}, \xi_t^{(2)}\big)$ with $\xi_t^{(g)} \geq 0$ and $\xi_t^{(1)} + \xi_t^{(2)} = 1$, splitting a fixed sampling effort $\kappa > 0$. We observe incident counts with Poisson noise:

$$y_t^{(g)} \,\big|\, \mathbf{X}_t, \boldsymbol{\theta}, \boldsymbol{\xi}_t \;\sim\; \mathrm{Poisson}\Big(\lambda_{t,(g)}^{\mathrm{obs}}\Big), \qquad \lambda_{t,(g)}^{\mathrm{obs}} \;=\; \kappa\,\xi_t^{(g)}\,\rho^{(g)}\,\frac{I_t^{(g)}}{N^{(g)}}, \quad g \in \{1, 2\}, \tag{17}$$

where $\rho^{(g)} > 0$ is a known detection scale.

Intuitively, the design vector $\boldsymbol{\xi}_t$ specifies how the fixed sampling effort $\kappa$ is distributed across groups, controlling the expected number of observations in each subpopulation. While it does not affect the underlying epidemic dynamics, it modulates the observation noise level and therefore the information gained about the unknown parameters.

**Regularity assumptions in this example.** The state projection in the Euler–Maruyama discretisation keeps the state in the feasible compact domain $S_t^{(g)}, I_t^{(g)} \in [0, N^{(g)}]$, with $S_t^{(g)} + I_t^{(g)} \leq N^{(g)}$. Together with bounded parameter priors and the compact design space, this restricts the particle system to a stable domain on which the transition rates in (14) depend smoothly on $\boldsymbol{\theta}$ and the state. This supports the stability condition in Assumption C.4 on the region explored by the particle approximation.

The Poisson observation model is strictly positive for all observed counts whenever the rate $\lambda_{t,(g)}^{\mathrm{obs}}$ is positive. In our experiments, the design constraints and simulated trajectories keep the relevant rates positive on the support used for inference, so the likelihood terms entering $L_{\boldsymbol{\theta},\boldsymbol{\xi}_t}(\boldsymbol{y}_t)$ and $Z_{\boldsymbol{\xi}_t}(\boldsymbol{y}_t)$ remain positive and the $\log(L/Z)$ integrand is well defined. Thus, while the boundedness and positivity conditions are satisfied on the effective support of the numerical experiment, providing the practical justification for Assumptions C.5 and C.6.

### E.2. Simulation Setup

We consider two subpopulations of equal size, $N^{(1)} = N^{(2)} = 200$, each initialized with $I_0^{(1)} = I_0^{(2)} = 5$ infectious individuals and $S_0^{(g)} = N^{(g)} - I_0^{(g)}$. The per-group detection scales are $\rho = \big(\rho^{(1)}, \rho^{(2)}\big) = (0.95, 0.5)$, and the total sampling effort is fixed to $\kappa = 100$. The cross-group mixing matrix is

$$\mathbf{M} = \begin{pmatrix} 0.9 & 0.1 \\ 0.1 & 0.9 \end{pmatrix},$$

which induces moderate within-group interaction and limited cross-group coupling. The continuous-time dynamics are integrated with an Euler–Maruyama step $\Delta\tau = 0.1$.

The true epidemiological parameters are $\beta^{(1)} = 0.65$, $\gamma^{(1)} = 0.15$ (for group 1, treated as unknown) and $\beta^{(2)} = 0.55$, $\gamma^{(2)} = 0.15$ (for group 2, fixed and known). We infer only $\boldsymbol{\theta} = \big(\beta^{(1)}, \gamma^{(1)}\big)$ with uniform priors

$$\beta^{(1)}, \gamma^{(1)} \sim \mathcal{U}(0.1, 1.0).$$

**Algorithmic settings.** For the NPF, we use $M = N = 100$ particles for parameters and states, respectively ($L = M \times N$ total Monte Carlo samples). We constrained the particle counts to $M = N$ and selected this value via grid search over $\{50, 100, 200, 300, 400, 500\}$, choosing the smallest particle count for which performance gains became marginal relative to the added computational cost. Parameter jittering is $\kappa_M(\cdot \,|\, \boldsymbol{\theta}') = \mathcal{N}(\boldsymbol{\theta} \,|\, \boldsymbol{\theta}', \sigma_{\mathrm{jitter}}^2 \boldsymbol{I}_{d_\theta})$, with variance scaled as

$$\sigma_{\mathrm{jitter}}^2 = \frac{c_{\mathrm{jitter}}}{M^{3/2}}.$$

*Table 2.* RNMSE results for the two-group SIR model. Values are bootstrap means with 95% confidence intervals.

*(a)* RNMSE for state estimation, $\mathrm{RNMSE}_x$.

| $t$ | Oracle | BAD-PODS | Random | Static |
|---|---|---|---|---|
| 50 | 0.058 | **0.057** | 0.089 | 0.061 |
| | [0.050, 0.067] | [0.050, 0.065] | [0.078, 0.101] | [0.054, 0.068] |
| 100 | 0.132 | 0.144 | 0.159 | **0.115** |
| | [0.119, 0.144] | [0.130, 0.158] | [0.143, 0.174] | [0.104, 0.127] |
| 150 | 0.133 | 0.136 | 0.166 | **0.111** |
| | [0.124, 0.144] | [0.123, 0.149] | [0.153, 0.181] | [0.102, 0.121] |
| 200 | 0.160 | **0.165** | 0.218 | – |
| | [0.146, 0.175] | [0.149, 0.183] | [0.195, 0.245] | |

*(b)* RNMSE for parameter inference, $\mathrm{RNMSE}_\theta$.

| $t$ | Oracle | BAD-PODS | Random | Static |
|---|---|---|---|---|
| 50 | 0.480 | 0.480 | 0.555 | **0.477** |
| | [0.424, 0.555] | [0.424, 0.557] | [0.497, 0.627] | [0.423, 0.542] |
| 100 | 0.275 | 0.287 | 0.306 | **0.211** |
| | [0.252, 0.299] | [0.266, 0.309] | [0.286, 0.324] | [0.190, 0.234] |
| 150 | 0.253 | 0.262 | 0.268 | **0.204** |
| | [0.236, 0.273] | [0.241, 0.286] | [0.248, 0.288] | [0.181, 0.227] |
| 200 | 0.308 | **0.279** | 0.319 | – |
| | [0.289, 0.330] | [0.256, 0.304] | [0.293, 0.348] | |

The scale constant $c_{\mathrm{jitter}}$ was tuned by a two-stage grid search: first a coarse grid $\{0.1, 0.5, 1, 5, 10\}$, followed by a refined grid $\{1, 2, 3, 4, 5\}$ around the best-performing region. We set $c_{\mathrm{jitter}} = 2$. We resample parameter particles at every time step in all experiments, although using an effective sample size (ESS) threshold to trigger resampling is also compatible with this algorithm.

In the BAD-PODS implementation, design optimisation at each step uses stochastic gradient ascent with $K = 500$ updates, where $K$ was selected via grid search over $\{50, 100, 200, 300, 400, 500, 600\}$. We use ADAM (Kingma & Ba, 2015) with standard momentum parameters $\beta_1 = 0.9$, $\beta_2 = 0.999$, and $\varepsilon = 10^{-6}$, and we tune the step size by grid search over $\alpha \in \{5 \times 10^{-4}, 10^{-3}, 2.5 \times 10^{-3}, 5 \times 10^{-3}, 10^{-2}, 2 \times 10^{-2}, 3 \times 10^{-2}, 4 \times 10^{-2}, 5 \times 10^{-2}, 7.5 \times 10^{-2}\}$, selecting $\alpha = 0.03$. Design variables are initialised uniformly, $\xi_t^{(1)} \sim \mathcal{U}(0, 1)$ and $\xi_t^{(2)} = 1 - \xi_t^{(1)}$, and mapped through a sigmoid reparameterisation to enforce simplex constraints. For the oracle implementation, instead of stochastic optimisation we do a grid search. We discretise $\xi_t^{(1)} \in [0, 1]$ into 500 candidate values, to obtain a fine grid resolution and select that design that maximises the EIG.

All experiments are repeated for $T = 200$ sequential design steps and averaged over 50 Monte Carlo realizations with independent random seeds.

### E.3. Additional Results for SIR Model

To complement the EIG-based evaluation, we also assess the quality of the resulting posterior inference. Tables 2a and 2b report the root normalised mean squared error (RNMSE) for the latent states and static parameters, respectively, computed against the ground-truth values used to simulate the data.

For the horizons where the static baseline can be run[2], it often yields the lowest RNMSE for both latent states and parameters. This indicates that the offline full-horizon design is strong for the downstream inference task in this susceptible-infectious-recovered (SIR) experiment. Among the sequential methods, the results mirror the EIG-based evaluation: BAD-PODS improves over random design and remains close to the oracle baseline.

## F. Moving Source Location Model

### F.1. Model Description

**State and parameters.** The latent state is $\boldsymbol{x}_t = (p_{x,t}, p_{y,t}, \phi_t)^\top \in \mathbb{R}^2 \times (-\pi, \pi]$, where $\boldsymbol{p}_t = (p_{x,t}, p_{y,t})^\top$ is position in a plane and $\phi_t$ heading angle. Static motion parameters are $\boldsymbol{\theta} = (v_x, v_y, v_\phi)$.

**Dynamics (transition).** With sampling step $\Delta t > 0$,

$$\boldsymbol{x}_t = \boldsymbol{x}_{t-1} + \Delta t \begin{bmatrix} v_x \cos \phi_{t-1} \\ v_y \sin \phi_{t-1} \\ v_\phi \end{bmatrix} + \boldsymbol{\epsilon}_t, \qquad \boldsymbol{\epsilon}_t \sim \mathcal{N}(\boldsymbol{0}, \mathbf{Q}), \tag{18}$$

with $\mathbf{Q} = \mathrm{diag}(\sigma_x^2, \sigma_y^2, \sigma_\phi^2)$. After propagation the heading is wrapped to the principal interval, $\phi_t \leftarrow \mathrm{wrap}_\pi(\phi_t)$, where $\mathrm{wrap}_\pi(\phi) = ((\phi + \pi) \mod 2\pi) - \pi$. The transition (18) does not depend on the design.

---

[2]The static baseline is not reported at longer horizons due to memory constraints.

**Sensors and design.** There are $J$ fixed sensors at positions $\{s_j\}_{j=1}^J \subset \mathbb{R}^2$. At time $t$ the design is the vector of orientations $\boldsymbol{\xi}_t = (\xi_{t,1}, \ldots, \xi_{t,J}) \in [-\pi, \pi)^J$. We define the bearing from sensor $j$ to the source as $\psi_{t,j}(\boldsymbol{p}_t) = \mathrm{atan2}\big((\boldsymbol{p}_t - \boldsymbol{s}_j)_y, (\boldsymbol{p}_t - \boldsymbol{s}_j)_x\big)$, and the angular mismatch $\Delta_{t,j} = \xi_{t,j} - \psi_{t,j}(\boldsymbol{p}_t)$.

**Observation model.** Each sensor reports a log–intensity corrupted by independent and identically distributed (i.i.d.) Gaussian noise,

$$\log y_{t,j} \,|\, \boldsymbol{p}_t, \boldsymbol{\theta}, \boldsymbol{\xi}_t \sim \mathcal{N}\big(\log \mu_{t,j}, \sigma^2\big), \tag{19}$$

$$\mu_{t,j} = b + \frac{\alpha_j}{m + \|\boldsymbol{p}_t - \boldsymbol{s}_j\|^2} \, D\big(\Delta_{t,j}\big), \tag{20}$$

for $j = 1, \ldots, J$, where $b > 0$ is a background level, $m > 0$ a saturation constant, $\alpha_j \geq 0$ a per-sensor strength, and

$$D(\delta) = \left(\frac{1 + d\cos\delta}{1 + d}\right)^k, \qquad d \in [0,1), \quad k > 1,$$

is a cardioid directivity function that favors alignment between the sensor orientation and the bearing to the source, while ensuring that $\mu_{t,j} > 0$ for all configurations. The design vector $\boldsymbol{\xi}_t$ enters the observation model (20) only through the angular offset $\Delta_{t,j}$.

**Regularity assumptions in this example.** The transition model in (18) is smooth in the static parameters $\boldsymbol{\theta} = (v_x, v_y, v_\phi)$ and in the state, interpreting the heading component as an angle on the circle. The wrapping operation simply maps equivalent angles back to the principal interval $(-\pi, \pi]$ and does not change the physical state. Since the parameter priors and design space are bounded in the experiments, the particle system evolves on the relevant support induced by the prior, dynamics, and observations. On this support, the transition and observation mappings depend continuously on $\boldsymbol{\theta}$, supporting the stability condition in Assumption C.4.

The observation likelihood is Gaussian in $\log y_{t,j}$ with variance $\sigma^2 > 0$. Moreover, the mean intensity satisfies $\mu_{t,j} > 0$ for all configurations because $b > 0$, $m > 0$, and the directivity function is non-negative and bounded. Thus, on the effective support explored in the experiments, the likelihood is strictly positive, smooth, and bounded for the realised observations. This keeps the likelihood terms entering $L_{\boldsymbol{\theta}, \boldsymbol{\xi}_t}(\boldsymbol{y}_t)$ and $Z_{\boldsymbol{\xi}_t}(\boldsymbol{y}_t)$ positive and well behaved, so that the $\log(L/Z)$ integrand is well defined and square-integrable, as required by Assumptions C.5 and C.6.

### F.2. Simulation Setup

We use $J = 2$ fixed sensors at $\boldsymbol{s}_1 = (3,0)^\top$ and $\boldsymbol{s}_2 = (0,3)^\top$. Per–sensor strengths are $\alpha_j = 5$ (for $j = 1, 2$), with background level $b = 0.1$ and saturation constant $m = 0.1$. The angular directivity $D(\delta)$ is set with $d = 1$ and $k = 4$. Observations are log–intensities with i.i.d. Gaussian noise $\sigma^2 = 0.1$, hence $R = \sigma^2 \mathbf{I}_J$. Designs are initialized uniformly as $\boldsymbol{\xi}_t \sim \mathcal{U}([-\pi, \pi))^J$.

For the state transition we use step $\Delta t = 0.1$ and process noise $\boldsymbol{\epsilon}_t \sim \mathcal{N}(\mathbf{0}, Q)$ with $Q = \mathrm{diag}(0.2, 0.2, 10^{-2})$. We infer only the planar velocity components $\boldsymbol{\theta} = (v_x, v_y)$, with uniform priors $v_x, v_y \sim \mathcal{U}(0.5, 1.5)$. The true (data–generating) parameters are $v_x^\star = v_y^\star = 1.0$ (with $v_\phi = 0.3$ known).

**Algorithmic settings.** We use an NPF with $M = N = 300$ particles for parameters and states (total $L = M \times N$ Monte Carlo samples per design step). We constrained the particle counts to $M = N$ and selected this value via grid search over $\{50, 100, 200, 300, 400, 500\}$, choosing the smallest particle count for which performance gains became marginal relative to the added computational cost. Parameter jittering is $\kappa_M(\cdot \,|\, \boldsymbol{\theta}') = \mathcal{N}(\boldsymbol{\theta} \,|\, \boldsymbol{\theta}', \sigma_{\mathrm{jitter}}^2 \boldsymbol{I}_{d_\theta})$, with

$$\sigma_{\mathrm{jitter}}^2 = \frac{c_{\mathrm{jitter}}}{M^{3/2}}.$$

The scale constant $c_{\mathrm{jitter}}$ was tuned by a two-stage grid search: a coarse grid $\{0.1, 0.5, 1, 5, 10\}$, followed by a refined grid $\{0.05, 0.1, 0.15, 0.2, 0.25\}$ around the best-performing region; we set $c_{\mathrm{jitter}} = 0.15$. We resample parameter particles at every time step.

In the BAD-PODS implementation, design optimisation at each step uses stochastic gradient ascent with $K = 300$ updates, where $K$ was selected via grid search over $\{50, 100, 200, 300, 400, 500, 600\}$. We use ADAM (Kingma & Ba, 2015) with

*Table 3.* RNMSE results for the moving-source model. Values are bootstrap means with 95% confidence intervals.

*(a)* RNMSE for state estimation, $\text{RNMSE}_x$.

| $t$ | Oracle | BAD-PODS | Random | Static |
|---|---|---|---|---|
| 10 | 0.132 | **0.142** | 0.211 | 0.217 |
| | [0.114, 0.165] | [0.123, 0.172] | [0.179, 0.270] | [0.181, 0.293] |
| 20 | 0.097 | **0.110** | 0.123 | **0.110** |
| | [0.079, 0.143] | [0.088, 0.180] | [0.094, 0.223] | [0.092, 0.130] |
| 30 | 0.062 | **0.063** | 0.078 | – |
| | [0.054, 0.071] | [0.055, 0.073] | [0.068, 0.089] | |
| 40 | 0.058 | **0.057** | 0.077 | – |
| | [0.051, 0.065] | [0.049, 0.065] | [0.069, 0.087] | |
| 50 | 0.048 | **0.050** | 0.064 | – |
| | [0.043, 0.055] | [0.044, 0.057] | [0.058, 0.071] | |

*(b)* RNMSE for parameter inference, $\text{RNMSE}_\theta$.

| $t$ | Oracle | BAD-PODS | Random | Static |
|---|---|---|---|---|
| 10 | 0.096 | 0.094 | **0.084** | 0.111 |
| | [0.085, 0.108] | [0.079, 0.110] | [0.072, 0.098] | [0.096, 0.126] |
| 20 | 0.138 | **0.127** | 0.128 | 0.130 |
| | [0.124, 0.152] | [0.109, 0.146] | [0.111, 0.145] | [0.112, 0.148] |
| 30 | 0.156 | 0.148 | **0.146** | – |
| | [0.138, 0.172] | [0.124, 0.173] | [0.127, 0.164] | |
| 40 | 0.169 | **0.165** | 0.167 | – |
| | [0.151, 0.187] | [0.139, 0.194] | [0.147, 0.188] | |
| 50 | 0.169 | 0.170 | **0.169** | – |
| | [0.147, 0.190] | [0.142, 0.198] | [0.147, 0.193] | |

$\beta_1 = 0.9$, $\beta_2 = 0.999$, and $\varepsilon = 10^{-6}$, and we tune the step size over $\alpha \in \{5 \times 10^{-4}, 10^{-3}, 2.5 \times 10^{-3}, 5 \times 10^{-3}, 10^{-2}, 2 \times 10^{-2}\}$, selecting $\alpha = 0.01$. For the oracle implementation, we discretise the 2D design space $(-\pi, \pi] \times (-\pi, \pi]$ into a $25 \times 25$ grid (625 candidate values), selecting the one that maximises the EIG.

All experiments are repeated for $T = 50$ sequential design steps and averaged over 50 Monte Carlo realizations with independent random seeds.

### F.3. Additional Results for the Moving-Source Model

To complement the EIG results, we also assess the quality of the resulting posterior inference in the moving-source experiment. Tables 3a and 3b report the RNMSE for the latent states and static parameters, respectively, computed against the ground-truth values used to simulate the data.

The parameter RNMSE is similar across methods, suggesting that the fixed parameters are identifiable from the available observations under a range of design strategies. The latent-state RNMSE, however, shows clearer differences. Our method improves over random design and remains close to the oracle baseline. This reflects the nature of the moving-source experiment, where the target is joint state–parameter inference: BAD-PODS improves localisation of the time-varying latent state while maintaining comparable parameter error.

## G. Ecological Growth Model

### G.1. Model Description

We consider a stochastic logistic growth model with harvesting, commonly used in ecological population dynamics (Zhou et al., 2009). The latent state is the (scaled) population size $x_t \in \mathbb{R}_+$ at discrete time $t$. The design $\xi_t \in [0, 1]$ controls harvesting intensity (e.g., fishing effort or trapping rate), and therefore affects both the transition dynamics and the observation model.

**Dynamics (transition).** Let $\Delta > 0$ be the integration step size. Conditioned on $(x_t, \xi_t)$ and parameters $\boldsymbol{\theta} = (r, k)$, the dynamics are

$$x_{t+1} = x_t + \Delta\left(r\, x_t\left(1 - \frac{x_t}{k}\right) - q\, \xi_t\, x_t\right) + \varepsilon_t, \qquad \varepsilon_t \sim \mathcal{N}(0, \Delta\sigma_x^2), \tag{21}$$

where $r > 0$ is the intrinsic growth rate, $k > 0$ is the carrying capacity, and $q > 0$ is the catchability coefficient. The term $q\, \xi_t\, x_t$ is the (expected) harvested amount per unit time, proportional to both effort $\xi_t$ and available population $x_t$.

**Observation model and design.** We define the harvested amount

$$\lambda_t = q\, \xi_t\, x_t. \tag{22}$$

Observations provide a noisy, partially saturated measurement of this harvested amount:

$$y_t \,|\, x_t, \xi_t, \boldsymbol{\theta} \sim \mathcal{N}\big(h(\lambda_t),\, \sigma_y^2\big), \qquad h(\lambda_t) = C_{\max} \frac{\lambda_t}{C_{\text{half}} + \lambda_t}, \tag{23}$$

where $C_{\max} > 0$ is the saturation level and $C_{\text{half}} > 0$ controls the transition from a linear to a saturated regime. This nonlinear response implies that increasing harvesting effort does not increase information indefinitely: for large $\lambda_t$ the function $h(\lambda_t)$ saturates, obtaining virtually same outcomes for different harvesting efforts.

The design is one-dimensional, $\xi_t \in [0, 1]$. Small $\xi_t$ yields weak signals $\lambda_t$ and therefore low signal-to-noise, while large $\xi_t$ can quickly depress the latent state $x_t$ through the transition (21) and also pushes $h(\lambda_t)$ into a saturation regime. Together, these effects create a non-trivial trade-off and can yield intermediate optimal designs.

**Regularity assumptions in this example.** The transition map in (21) depends smoothly on the parameters $\boldsymbol{\theta} = (r, k)$, the state $x_t$, and the design $\xi_t$, provided $k$ is bounded away from zero. In our experiments, the parameter priors are bounded with $r > 0$ and $k > 0$, and the design space is compact, $\xi_t \in [0, 1]$. The particle system therefore evolves on the effective support induced by the prior, dynamics, and observations, where the transition and observation mappings are continuous in $\boldsymbol{\theta}$. This supports the stability condition in Assumption C.4.

The observation model is Gaussian with variance $\sigma_y^2 > 0$, and the observation function $h(\lambda_t)$ is smooth and bounded for $\lambda_t \geq 0$ because $C_{\max} > 0$ and $C_{\text{half}} > 0$. Thus, on the effective support explored in the experiments, the likelihood is strictly positive, smooth, and bounded for the realised observations. This keeps the likelihood terms entering $L_{\boldsymbol{\theta}, \boldsymbol{\xi}_t}(\boldsymbol{y}_t)$ and $Z_{\boldsymbol{\xi}_t}(\boldsymbol{y}_t)$ positive and well behaved, so that the $\log(L/Z)$ integrand is well defined and square-integrable, as required by Assumptions C.5 and C.6.

## G.2. Simulation Setup

We simulate the growth/harvest model in (21)–(23) over a horizon of $T = 20$ steps, with integration step $\Delta = 0.1$. The initial state is set to $x_0 = 0.4\,k$. The true (data-generating) parameters are $r^\star = 0.5$ and $k^\star = 300$. We assume process noise variance $\sigma_x^2 = 0.1$, observation noise variance $\sigma_y^2 = 2.0$, and saturation parameters $C_{\max} = 90$ and $C_{\text{half}} = 30$ are known. The design is the harvest effort $\xi_t \in [0, 1]$, initialised at each time step as $\xi_t \sim \mathcal{U}(0, 1)$. We infer only $\boldsymbol{\theta} = (r, k)$, with independent priors $r \sim \mathcal{U}(0.2, 1.2)$ and $k \sim \mathcal{U}(200, 800)$.

**Algorithmic settings.** We use an NPF with $M = N = 200$ particles for parameters and states (total $L = M \times N$ Monte Carlo samples per design step). We constrained the particle counts to $M = N$ and selected this value via grid search over $\{50, 100, 200, 300, 400, 500\}$, choosing the smallest particle count for which performance improvements became marginal relative to the added computational cost. Parameter jittering is $\kappa_M(\cdot \,|\, \boldsymbol{\theta}') = \mathcal{N}(\boldsymbol{\theta} \,|\, \boldsymbol{\theta}', \Sigma_{\text{jitter}})$, with

$$\Sigma_{\text{jitter}} = \begin{bmatrix} \sigma_r^2 & 0 \\ 0 & \sigma_k^2 \end{bmatrix}, \qquad \sigma_r^2 = \frac{c_r}{M^{3/2}}, \qquad \sigma_k^2 = \frac{c_k}{M^{3/2}}.$$

The scale constants $c_r$ and $c_k$ were tuned by a two-stage grid search: a coarse grid $\{0.1, 0.5, 1, 5, 10\}$ and $\{1, 5, 10, 20, 40, 80\}$, followed by a refined grid around the best-performing region. We set $c_r = 0.05$ and $c_k = 50$. We resample parameter particles at every time step.

In the BAD-PODS implementation, design optimisation at each step uses stochastic gradient ascent with $K = 200$ updates, where $K$ was selected via grid search over $\{50, 100, 200, 300, 400, 500, 600\}$. We use ADAM (Kingma & Ba, 2015) with $\beta_1 = 0.9$, $\beta_2 = 0.999$, and $\varepsilon = 10^{-6}$, and we tune the step size over $\alpha \in \{5 \times 10^{-4}, 10^{-3}, 2.5 \times 10^{-3}, 5 \times 10^{-3}, 10^{-2}, 2 \times 10^{-2}, 3 \times 10^{-2}, 4 \times 10^{-2}, 5 \times 10^{-2}\}$, selecting $\alpha = 5 \times 10^{-3}$. For the oracle baseline, we discretise the 1D design space $[0, 1]$ into a grid of 500 candidates and select $\xi_t$ by maximising the estimated EIG over this grid at each time step.

All experiments are averaged over 50 Monte Carlo realisations with independent random seeds.

## G.3. Additional Results for the Ecological Growth Model

To complement the EIG-based evaluation, we also assess the quality of the resulting posterior inference in the ecological growth experiment. Tables 4a and 4b report the RNMSE for the latent states and static parameters, respectively, computed against the ground-truth values used to simulate the data.

For latent-state inference, BAD-PODS achieves low RNMSE across horizons, remaining close to the oracle baseline and improving over random design in mean error. The static baseline is competitive at the shortest horizons, but is only available for $t \leq 10$ in this experiment (due to computation and memory constraints). For parameter inference, the differences are

*Table 4.* RNMSE results for the ecological growth model. Values are bootstrap means with 95% confidence intervals.

*(a)* RNMSE for state estimation, $\text{RNMSE}_x$.

| $t$ | Oracle | BAD-PODS | Random | Static |
|---|---|---|---|---|
| 5 | 0.029 | 0.025 | 0.035 | **0.024** |
| | [0.023, 0.036] | [0.018, 0.035] | [0.028, 0.045] | [0.019, 0.030] |
| 10 | 0.028 | **0.027** | 0.032 | 0.029 |
| | [0.022, 0.034] | [0.022, 0.034] | [0.025, 0.041] | [0.024, 0.036] |
| 15 | 0.023 | **0.027** | 0.032 | – |
| | [0.019, 0.028] | [0.022, 0.037] | [0.025, 0.040] | |
| 20 | 0.024 | **0.026** | 0.032 | – |
| | [0.020, 0.029] | [0.020, 0.036] | [0.026, 0.040] | |

*(b)* RNMSE for parameter inference, $\text{RNMSE}_\theta$.

| $t$ | Oracle | BAD-PODS | Random | Static |
|---|---|---|---|---|
| 5 | 0.494 | **0.535** | 0.558 | 0.552 |
| | [0.461, 0.531] | [0.487, 0.577] | [0.514, 0.608] | [0.508, 0.597] |
| 10 | 0.512 | **0.533** | 0.581 | **0.533** |
| | [0.456, 0.571] | [0.462, 0.604] | [0.529, 0.637] | [0.486, 0.584] |
| 15 | 0.488 | **0.477** | 0.606 | – |
| | [0.412, 0.572] | [0.384, 0.580] | [0.531, 0.679] | |
| 20 | 0.464 | **0.524** | 0.613 | – |
| | [0.377, 0.571] | [0.421, 0.636] | [0.522, 0.704] | |

more evident, where BAD-PODS shows a clear reduction in mean RNMSE relative to random design, while remaining competitive with the oracle baseline.

## H. Computational Infrastructure

All experiments were conducted on a high-performance computing (HPC) cluster equipped with NVIDIA V100 GPUs. The computations made use of multi-core Intel Xeon processors, high-capacity DDR4 memory, and InfiniBand interconnects for fast data transfer between nodes. Each GPU node contained multiple V100 devices (16–32 GB memory per GPU), and all optimisation and inference workloads were executed using these resources.

---

**Algorithm 1** BAD-PODS: Bayesian adaptive design for partially observable dynamical systems

---

**Require:** Particle counts $(M, N)$; inner stochastic gradient ascent (SGA) steps $K$ with stepsizes $\{\eta_k\}_{k=0}^{K-1}$; priors $p(\boldsymbol{\theta})$, $p(\boldsymbol{x}_0)$

1: **Initialization:** $\boldsymbol{\theta}_0^{(m)} \sim p(\boldsymbol{\theta})$, $\boldsymbol{x}_0^{(m,n)} \sim p(\boldsymbol{x}_0)$; $w_{\boldsymbol{\theta},0}^{(m)} = 1/M$, $w_{\boldsymbol{x},0}^{(m,n)} = 1/N$

2: **for** $t = 1$ to $T$ **do**

3:     Initialize $\boldsymbol{\xi}_t^{(0)}$ randomly                                                                     $\triangleright$ *design optimisation loop*

4:     **for** $k = 0$ to $K - 1$ **do**

5:         Draw outer MC samples $(\widetilde{\boldsymbol{y}}_t^{(\ell)}, \widetilde{\boldsymbol{x}}_t^{(\ell)}, \boldsymbol{\theta}_{t-1}^{(\ell)}) \sim \Gamma(\cdot \,|\, \boldsymbol{\xi}_t^{(k)}, h_{t-1})$ (Appendix B)

6:         For each $m$, sample $\ddot{\boldsymbol{x}}_t^{(m,n)} \sim f(\cdot \,|\, \boldsymbol{x}_{t-1}^{(m,n)}, \boldsymbol{\theta}_{t-1}^{(m)}, \boldsymbol{\xi}_t^{(k)})$ for $n = 1{:}N$           $\triangleright$ *for* $\widehat{L}$

7:         Jitter $\dot{\boldsymbol{\theta}}_{t-1}^{(m)} \sim \kappa_M(\cdot \,|\, \boldsymbol{\theta}_{t-1}^{(m)})$ and sample $\dot{\boldsymbol{x}}_t^{(m,n)} \sim f(\cdot \,|\, \boldsymbol{x}_{t-1}^{(m,n)}, \dot{\boldsymbol{\theta}}_{t-1}^{(m)}, \boldsymbol{\xi}_t^{(k)})$         $\triangleright$ *for* $\widehat{Z}$

8:         Compute $\widehat{L}_{\boldsymbol{\theta}_{t-1}, \boldsymbol{\xi}_t^{(k)}}^{N}$ and $\widehat{Z}_{\boldsymbol{\xi}_t^{(k)}}^{M,N}$ in (7)–(8) in Section 3.3

9:         Compute $\widehat{\nabla_{\boldsymbol{\xi}_t} \mathcal{I}}(\boldsymbol{\xi}_t^{(k)})$ in (10) in Appendix B

10:        $\boldsymbol{\xi}_t^{(k+1)} \leftarrow \boldsymbol{\xi}_t^{(k)} + \eta_k \widehat{\nabla_{\boldsymbol{\xi}_t} \mathcal{I}}(\boldsymbol{\xi}_t^{(k)})$

11:     **end for**

12:     $\boldsymbol{\xi}_t^{\star} \leftarrow \boldsymbol{\xi}_t^{(K)}$; collect $\boldsymbol{y}_t \sim g(\cdot \,|\, \boldsymbol{\xi}_t^{\star})$                                   $\triangleright$ *execute design, collect data*

13:     Jitter parameters: $\boldsymbol{\theta}_{t|t-1}^{(m)} \sim \kappa_M(\cdot \,|\, \boldsymbol{\theta}_{t-1}^{(m)})$

14:     **for** $m = 1$ to $M$ **do**

15:         Propagate states: $\boldsymbol{x}_{t|t-1}^{(m,n)} \sim f(\cdot \,|\, \boldsymbol{x}_{t-1}^{(m,n)}, \boldsymbol{\theta}_{t|t-1}^{(m)}, \boldsymbol{\xi}_t^{\star})$

16:         Update state weights and normalize them

$$w_{\boldsymbol{x},t}^{(m,n)} \propto w_{\boldsymbol{x},t-1}^{(m,n)} \, g\Big(\boldsymbol{y}_t \,|\, \boldsymbol{x}_{t|t-1}^{(m,n)}, \boldsymbol{\theta}_{t|t-1}^{(m)}, \boldsymbol{\xi}_t^{\star}\Big), \qquad \widetilde{w}_{\boldsymbol{x},t}^{(m,n)} = \frac{w_{\boldsymbol{x},t}^{(m,n)}}{\sum_{j=1}^{N} w_{\boldsymbol{x},t}^{(m,j)}}$$

17:         Resample states: Draw indices $n_1, \ldots, n_N$ with probability $\widetilde{w}_{\boldsymbol{x},t}^{(m,1)}, \ldots, \widetilde{w}_{\boldsymbol{x},t}^{(m,N)}$

           Set $\boldsymbol{x}_t^{(m,n)} = \boldsymbol{x}_{t|t-1}^{(m,n_j)}$, for $j = 1 : N$                      $\triangleright$ *update conditional state posterior (NPF)*

18:     **end for**

19:     Update parameter weights and normalize them:

$$w_{\boldsymbol{\theta},t}^{(m)} = w_{\boldsymbol{\theta},t-1}^{(m)} \sum_{n=1}^{N} w_{\boldsymbol{x},t}^{(m,n)}, \qquad \widetilde{w}_{\boldsymbol{\theta},t}^{(m)} = \frac{w_{\boldsymbol{\theta},t}^{(m)}}{\sum_{i=1}^{M} w_{\boldsymbol{\theta},t}^{(i)}}$$

20:     Resample parameters: Draw indices $m_1, \ldots, m_M$ with probability $\widetilde{w}_{\boldsymbol{\theta},t}^{(1)}, \ldots, \widetilde{w}_{\boldsymbol{\theta},t}^{(M)}$

           Set $\boldsymbol{\theta}_t^{(m)} = \boldsymbol{\theta}_{t|t-1}^{(m_i)}$, for $i = 1 : M$                               $\triangleright$ *update parameter posterior (NPF)*

21: **end for**

---

