# OpenReview forum: "Online Bayesian Experimental Design for Partially Observed Dynamical Systems"
_ICML.cc/2026/Conference — ICML 2026 regular_

### Official Review · Reviewer_foGd · 2026-02-27

**Soundness:** 3
**Presentation:** 4
**Significance:** 2
**Originality:** 3
**Overall Recommendation:** 4
**Confidence:** 2

**Summary:**

This paper addresses the very challenging problem of online experiment design in state space models. That is, the goal is to take actions that produce the most informative future observations (given current uncertainty) at each time step. The general approach is to use a nested particle filter with jittering on the hierarchical variable to estimate the EIG gradient, and then use this to guide the action selection by stochastic gradient ascent to maximize expected information gain.

**Compliance With Llm Reviewing Policy:**

Affirmed.

**Key Questions For Authors:**

Do you think the nature of the jittering on the hierarchical variable matters?

How easy is it to determine if enough particles are being used?

Is there any natural connection to the body of work on amortized methods for static problems? Realistically, it seems like useful real-world examples are going to require some degree of amortization, or is that not a fair assessment?

**Limitations:**

I would have liked to see a more frank discussion of limitations of the method.

**Strengths And Weaknesses:**

To my understanding, the paper is sound. I have not carefully verified the central formulas such as (6).

In terms of significance, the paper treats a very challenging problem which to my knowledge has not been addressed before. The approach of using a nested particle filter with jittering on the hierarchical variable is not new, but as I understand it, the estimator of the EIG gradient is, and it is impressive that it gets any traction.

However, while the method performs well compared to various baselines, it is unclear to me whether it will prove useful in real applications. The examples, while nicely chosen, are all essentially toy cases. That said, given the novelty of the problem and approach, I don't think this is a major weakness, and I imagine this paper could be influential in the literature.

I found the presentation to be clear, at least as much as possible for a quite complicated problem. \ddot x notation is used several times. I was unsure whether I was meant to interpret this as a derivative, and this made following some of the technical details harder.

The experiments section presents three different cases, all of which nicely demonstrate the behavior of the method.

---

> ### Author Rebuttal · Authors · 2026-03-30
>
> We thank the reviewer for the careful reading and constructive feedback. We appreciate your assessment that the paper is sound, and that the presentation is “clear, at least as much as possible for a quite complicated problem.” We also appreciate that you acknowledge the problem as “very challenging,” highlighting the novelty of the EIG-gradient estimator. Finally, we thank you for the positive evaluation of the empirical section, noting that the three cases “nicely demonstrate the behaviour of the method,” and for your perspective that, despite being toy problems, the work could still be “influential in the literature.” We address each of your comments below.
>
>  1. **Jittering adjustment.** Thank you for raising this point. We have added explicit discussion and practical guidance on selecting the jittering variance. The jittering scale matters: in an NPF, it is not merely a generic rejuvenation step, but it controls both (i) exploration of the parameter space and (ii) stability of the inner conditional state filters. If the variance is too small, parameter exploration is slow, and weights can collapse onto a few particles; if it is too large, the parameter move becomes inconsistent with the conditional state approximations, which can accumulate error and also lead to degeneracy. Since the appropriate scale depends on the model and on the parameter units, it is not obvious to set a priori. We now discuss this sensitivity and tuning issue explicitly with other limitations of the approach.
>
>  2. **Number of particles.** We also added guidance on diagnosing whether the particle counts are sufficient. While increasing $(M,N)$ improves accuracy (at higher runtime), inadequate counts typically manifest through weight degeneracy. In practice, we can monitor effective sample size (ESS) at both layers (parameter and state), where persistently low ESS indicates insufficient $(M,N)$ and/or poorly tuned jittering. As an additional offline check, we can rerun a subset of experiments with increasing $(M,N)$ and verify that the selected designs and EIG curves stabilise.
>
>  3. **Limitations and connection to amortised methods.** We have expanded the limitations discussion and clarified the relation to amortised approaches.
> **BAD-PODS removes the growth of cost with time** by keeping inference and design updates online (stable per-step cost), making long horizons feasible. However, it can still be computationally demanding in challenging partially observed settings because it relies on Monte Carlo sampling: **the per-step cost increases with the particle counts $(M,N)$ and the number of optimisation steps $K$**, and the required number of particles can grow rapidly with state/parameter dimension.
> **Amortised methods, in contrast, enable very fast deployment once a policy is trained,** which is attractive under strict real-time constraints. Note that in our setting, **a direct adaptation of amortised methods is not straightforward,** as it requires changing the training objective and estimators to accommodate an intractable likelihood. But even if amortised methods are adapted to partially observed SSMs, **they may struggle in long-horizon settings due to coverage and generalisation:** training must approximate a policy over an expanding history space, and for large state spaces (e.g., moving-source trajectories that diverge substantially from similar initial conditions), adequate coverage becomes prohibitive. The step-DAD line of work (Hedman et al., 2025) explicitly motivates intermediate retraining to reduce suboptimality, indicating that “one-shot” amortisation may not remain optimal as experiments progress. We now make these points more explicit in the manuscript.
>
> Once again, thank you for all the comments. We believe the revisions address the points you raised, and we would be glad if you could take into account these changes when reconsidering the score. **We are eager to provide additional clarifications if any concerns still remain.**

---

> > ### Author Rebuttal · Reviewer_foGd · 2026-04-02
> >
> > Thanks - this answers my questions.

---

> > > ### Author Response · Authors · 2026-04-03
> > >
> > > Thank you for engaging. Please do let us know of any action or clarification that you find necessary to increase the score.

---

### Official Review · Reviewer_t4FZ · 2026-03-06

**Soundness:** 2
**Presentation:** 2
**Significance:** 2
**Originality:** 2
**Overall Recommendation:** 3
**Confidence:** 5

**Summary:**

This paper addresses sequential Bayesian optimal experimental design in settings where observations are the result of an unobserved latent dynamical system. The likelihood is not directly available and requires integration over all latent variables which themselves follow a Markov chain. To face this difficulty, the authors propose to use nested particle filters to provide likelihood and marginal approximations that they plug then in standard EIG-based BOED.

**Compliance With Llm Reviewing Policy:**

Affirmed.

**Final Justification:**

The paper is interesting but would require more work to reach the demanding level of the conference. It's obvious to me that some parts have been rushed through, like a somewhat careless writing of formulas and clear formulation of made assumptions, simplistic evaluation which essentially reflects lack of time before submission, etc. There remains a number of questions to be treated with more care and I feel the paper would need a bit of additional work and is not fully ready for this round. The rebuttal didn't convinced me that the remaining questions were minor and I'm actually tempted to lower my score to "reject" but I keep it to "weak reject" as initially.

**Key Questions For Authors:**

See more detailed comments and questions above.
- Confirm and correct all mentioned typos/errors if relevant. Explain better otherwise.
- Better specify all conditional independence assumptions, and in particular to justify the missing term in $s$.
- Can the authors show that their assumptions are actually satisfied in their numerical examples?
-Justify the choice of their approximate EIG as an evaluation metric in the numerical section.
- Add a discussion on limitations
- Specify all running times in more details.

**Limitations:**

The technical limitations have not been discussed.

**Strengths And Weaknesses:**

## Soundness

**Reasonable assumptions?**
There is one theoretical consistency result in section 3.4, which is proved under a number of assumptions detailed in Appendix C. The proof seems to follow traditional schemes which is reassuring but can the authors show that their assumptions are actually satisfied in their numerical examples, in particular assumptions C4-6? A discussion checking the validity of these assumptions in the Appendix would be a good addition. If this is not possible, I would at least add a remark about this potential limitation. See also my concerns about the numerical evaluation below.

**Limitations of the work are not discussed.** In addition, the authors should at least comment on the scalability of their approach, which I suspect is limited to small dimensions due to its computational cost. Although NMC estimators have been studied in Rainforth et al 2018, it seems that they have been replaced in practice by other approaches since then.

**Running times** are only specified for their first numerical example and it seems that 12 minutes for each step is a lot. What is the total run time in this case? For the other experiments? The authors acknowledge the use of HPC for their experiments, which seems a bit overkilling considering the dimensions of their examples and add to the feeling that the method is extremely costly.

**Other comments on the numerical evaluation:** In section 5, the method is said to be assessed with the Total EIG, which sounds weird as the Total EIG like the EIG is intractable. In the following lines, we can then understand from the notation that the authors are using their own approximation of the EIG to assess their method. This sounds a bit weak, as if their EIG approximation is bad, it may not show up in the experimental results. Especially, as they do not compare with any other method. I’m aware that this would require some work, but it does not seem impossible to re-use their approximation of the likelihood to plug in other methods, or to investigate methods like iDAD in their settings. The fact that their EIG estimator is consistent under some assumptions is not enough reassuring to me (see comment above) as we know that the value of this kind of results is often only theoretical as they are often not so close to what is numerically observed. The authors should at least check that their theoretical result applies.

## Presentation

The paper has a clear narrative and positioning. However, the technical part of the paper could be better presented or with more care. Starting from section 2 and the model presentation, a number of inconsistencies, typos or errors prevent from checking and following the derivations and thus assessing the validity of the approach. For such a selective venue, this is not acceptable. It suggests that the work is not ready for publication yet although it may contain interesting and useful ideas.
- First conditional independence assumptions should be better specified to avoid ambiguities.
- Section 2.1: definition of the sequential EIG and eq (1), $h_{t-1}$ is missing in $p(y_t | \xi_t)$ in (1). It follows that the second line of (1) should be corrected too.
Below (1) and in section 2.3, the likelihood is inconsistently denoted by $p(y_t | \theta, \xi_t, h_{t-1})$  or $p(y_t | \theta, \xi_t)$ in turn.
- Same remark for the marginal predictive, $h_{t-1}$ is missing in Section 3.1 line 3
- More importantly, it seems to me that $\xi_t$ is missing in the definition of Z in section 3.1. The expectation part should include $p(x_{0:t} | \theta, h_{t-1}, \xi_t)$.
This impacts also the definition of $\Gamma$ in section 3.2 as $\xi_t$ seems to be missing in the conditioning of the third term.
Why is it so? The authors should specify what kind of conditional independence assumption allows to remove $\xi_t$.
This seems important to me, as it has potential consequence on the definition of $s$ in section (6), possibly adding another intractable gradient term to compute, not handled in the current version of the method.
- Minor remark on the gradient in (6): The derivations of the gradient in (6) specified in Appendix A is unnecessarily lengthy. The same can be shown in about 3 lines.
- Equation (5) needs to be corrected too.
- Remaining (??) in Section G.2: this is very minor but adds to the feeling that the work needs a bit more proofreading before it’s ready.

## Significance

The setting addressed is useful in applications I believe. However, for now, I have doubts about the method performance in practice and I feel the paper is not yet ready for publication in this round.

## Originality

Moderate. What the authors are trying to do is useful but due to inconsistencies mentioned above and a lack of a proper discussion on limitations, I have doubt if the method is entirely valid. Also, the numerical assessment could be improved and more demonstrating. See comments above and questions below.

I will start with an assessment on the low side, as I feel the paper is lacking is several aspects. Of course, the work has some merits and I’m happy to reconsider this position as I may be wrong or too severe in my assessment.

---

> ### Author Rebuttal · Authors · 2026-03-30
>
> We thank the reviewer for the careful reading and constructive comments. We appreciate that you found the manuscript has “clear narrative and positioning,” and that “the setting addressed is useful in applications.” We also appreciate your willingness to reconsider your assessment as clarifications are incorporated. We address your comments below.
>
> 1. **Assumptions in the examples.**
> Thanks for the great suggestion to verify assumptions in our examples. We now clarify how C4–C6 hold on the *relevant support* induced by the priors, the dynamics, and design constraints. In the moving-source and growth models, observations are Gaussian with variance bounded away from 0, so $g(\cdot)$ is strictly positive, bounded, and smooth (C5). Combined with smooth transitions with additive Gaussian noise, this yields local Lipschitz/continuity of $L, Z$ estimators and square-integrability of $\log(L/Z)$ (C6). In the SIR model, observations are Poisson, then $g(\cdot)$ is strictly positive whenever the rate $\lambda$ is positive, which is the case in our setup (C5). This keeps $L, Z$ bounded away from zero and $\log(L/Z)$ well-defined (C6). C4 is supported by the smooth dependence of the transition/observation models on $\theta$ and by the state remaining in stable domains (e.g., $S^{(g)},I^{(g)}\in[0,N^{(g)}]$ in SIR).
>
>  2. **Evaluation with EIG/TEIG.**
> In most BED settings (partially observable or not), the EIG is doubly intractable, so **it is standard to evaluate designs using Monte Carlo EIG estimates (or bounds) under the assumed model.** For example, iDAD (Ivanova et al., 2021) reports accumulated/total EIG via sPCE (an approximate lower bound), and IO-SMC$^2$ (Iqbal et al., 2024) reports both EIG estimates and sPCE.
> The reviewer mentions "if their EIG approximation is bad, it may not show up in the experimental results", but we note that, **for our estimator, increasing particle counts will always improve the EIG approximation,** while approximate bounds do not provide the same type of consistency guarantee.
>
>  3. **Notation issues.**
> We thank the reviewer for taking the time to check the equations in detail. **We agree there were typos in the main text, and we have corrected them. We now state the conditional-independence structure more explicitly:** $\xi_t$ is chosen given $h_{t-1}$ and affects only the transition/observation at time $t$.
> In particular, in Section 3.1 terms like $p(x_{0:t}\mid \theta,h_{t-1})$ should read $p(x_{0:t}\mid \theta,\xi_t,h_{t-1})$. **This dependence was already handled correctly in the appendix derivations** (Appendix A), and was only missing in the main text.
> By contrast, in Section 3.2, $\Gamma_{\xi_t}$ correctly uses $p(x_{0:t-1}\mid \theta,h_{t-1})$ since past states are independent of the current design $\xi_t$, which instead enters the current-step factors $f(\cdot\mid \xi_t)$ and $g(\cdot\mid \xi_t)$.
> Finally, **we previously abused notation by omitting conditioning on $h_{t-1}$ in likelihood and evidence for brevity.** We now write the history dependence consistently, $p(y_t\mid \theta,\xi_t,h_{t-1})$ and $p(y_t\mid \xi_t,h_{t-1})$, to avoid any ambiguity.
>
> 4. **Limitations.**
> We have now added further limitations discussion. BAD-PODS avoids replaying the full history, yielding linear overall cost in $T$ (stable per-step complexity). However, the **per-step cost can still be substantial** because EIG/gradient estimation requires sampling latent-states and parameters, and achieving low-variance estimates may require large $(M,N)$. **Scalability with state/parameter dimension remains challenging,** as particle methods typically need increasing $M,N$ to maintain accuracy. However, **while competitor methods may run faster, their approximation error is not easy to quantify and will also typically grow with state/parameter dimension.** Performance also depends on tuning---especially the jittering variance, which trades off exploration of $\theta$ against stability of the conditional filters.
>
> 5. **Runtime and memory.**
> We now report wall-clock runtimes and memory usage for all experiments. For the **growth** model ($T=10$), runtimes are $\approx$32s (random), 311 min (static), and 25 min (BAD-PODS); memory is $\approx$25GB GPU /2GB CPU (static) vs. $\approx$13GB GPU/1GB CPU (BAD-PODS). For the **moving-source** model ($T=20$), runtimes are $\approx$2 min (random), 628 min (static), and 525 min (BAD-PODS); memory is $\approx$15GB GPU/5GB CPU (static) vs. $\approx$13GB GPU/1GB CPU (BAD-PODS). As in SIR, static BED scales poorly compared to our approach. However, runtime in BAD-PODS increases in harder settings where stable inference requires larger $M,N$ (as in moving-source).
>
> Once again, thank you for the useful comments. We hope the revisions address the issues you raised, and we would be glad if you could take into account these changes when reconsidering the score. **We are eager to provide additional clarifications if any concerns still remain.**

---

> > ### Author Rebuttal · Reviewer_t4FZ · 2026-04-02
> >
> > Thanks for your reply. My main concerns remain regarding your answer in point 3. I'm still not convinced that your expression of $\Gamma_{\xi_t}$ in section 3.2 is correct. You claim that $x_{0:t-1}$ is independent of $\xi_t$ conditionally to $\theta$ and $h_{t-1}$ but it seems to me that this is not true and the purpose of my comments was to ask for more details. I'm sorry that this important point was mixed with less important typos.
> >
> >  Indeed, $x_{0:t-1}$ impact $x_t$ which in turn impacts $y_t$ which itself depends on $\xi_t$ or vice-versa. So even with the conditioning, there remains a dependence through $x_t$ and $y_t$. In other words, using a graphical model representation of your model, it seems to me that $\xi_t$ and $x_{0:t-1}$ remain connected on the graph through $x_t$ and $y_t$, even conditionally on $\theta$ and $h_{t-1}$.
> >
> > So my conclusion for now is that the 3rd term in your $\Gamma_{\xi_t}$ misses $\xi_t$ and I thus still have serious doubts about the validity of (6) and the tractability of your procedure should the correction I mentioned be made.
> >
> > I'm also not convinced by your evaluation with your approximated EIG, a more neutral metric would have been preferable.
> >
> > In addition, I also read the comments of the other reviewers, and it seems to me that other imperfections remain. Considering the standing of the conference, I feel the overall score is already too high for a paper which is not yet fully ready for publication. I however keep my score as is.
> >
> > Additional comment after the reply by the authors:
> >
> > - thanks for specifying your point about filtering and smoothing but I still have doubts. To me smoothing and filtering are just two possible tasks when dealing with state-space models. The fact that you target one or the other task should not change the dependence structure in the model. And we are not talking here about looking in the far future but just reading your state-space equations.
> >
> > - Regarding evaluation metrics, I was thinking about the usual SPCE and SNMC (Foster et al 2021) which have the advantage of depending only on the design sequences. But that said, the main problem in your evaluation is that you only compare with random design, which is not the most difficult procedure to outperform. So I suspect that whatever the metric, your method should do better than random, I hope.
> >
> > I'm sorry if I'm missing your point but I thus still need to be convinced regarding the first issue. I would suggest you write all your assumptions clearly in terms of probability distributions (conditional independencies and all) and then show using the involved pdfs the conditional independence property I'm questioning.

---

> > > ### Author Response · Authors · 2026-04-03
> > >
> > > ### On $\Gamma_{\xi_t}$ and conditioning on $\xi_t$.
> > >
> > > Thank you for pushing on this point. You are right that, in some settings, the belief about past states can be conditioned on future observations/designs—this is the **smoothing** perspective in inference ([1], Chapter 12). Concretely, given a full rollout $h_T=(\xi_{1:T},y_{1:T})$, one can form $p(x_{0:t}\mid \theta, h_T)$, which differs from the **filtering** perspective ([1], Chapter 6). Thus, it is not correct to state that $x_{0:t-1}$ is conditionally independent of information at time $t$ (like $\xi_t$ or $y_t$). However, **we are not in the smoothing setting**. We are in the context of **online decision-making**, thus, we work with filtering. Naturally, this is online, as also stated in the title of the paper. Conditioning on future history is possible in a smoothing (offline) setting, but it is not our goal. Therefore, **the equations match our intended setting and our methodology remains valid.**
> > >
> > > ### On evaluation metrics.
> > >
> > > **We now include complementary metrics** in addition to TEIG: (i) parameter RNMSE and (ii) state RNMSE w.r.t. ground truth (see the new tables added in response to Reviewer e51L). These results confirm that higher TEIG corresponds to improved inference quality rather than being an artefact of the estimator. We still report EIG/TEIG because it directly matches the optimisation objective and is standard in related work (e.g., iDAD in [2]; IO-SMC$^2$ in [3]). **If the reviewer prefers some other additional metrics, we would be happy to include them.**
> > >
> > > [1] Sarkkä and Svensson (2023). Bayesian filtering and smoothing. Cambridge University Press, 2nd edition.
> > >
> > > [2] Ivanova et al. (2021). Implicit deep adaptive design: Policy-based experimental design without likelihoods. Advances in neural information processing systems, 34, 25785-25798.
> > >
> > > [3] Iqbal et al. (2024). Nesting particle filters for experimental design in dynamical systems. In Proceedings of International Conference on Machine Learning (ICML).

---

### Official Review · Reviewer_q42P · 2026-03-07

**Soundness:** 4
**Presentation:** 3
**Significance:** 2
**Originality:** 2
**Overall Recommendation:** 4
**Confidence:** 3

**Summary:**

Dynamical Systems are ubiquitous across all areas of science and engineering. Increasing the efficiency of data collection process would thus be of significant impact. This paper proposes and online Bayesian experimental design method for partially observable systems with fully online inference. Technically the paper is very good and nice to read. The problem would be of significant practical impact if solved.

**Compliance With Llm Reviewing Policy:**

Affirmed.

**Final Justification:**

Overall I am conflcted about the paper. I really thing the overall direction and motivation is very important and that the paper is technically sound. My main concerns are the lack of comparison with amortized approaches as well as a limited demonstration of scale and clear benfits over baselines in the experiments. I would thus place it as a weak accept but very happy if other reviewers convince me that this should be accepted.

**Key Questions For Authors:**

Can you include an amortized baseline, is the nested particle approach really more scalable or compute efficient than amortization, can you actually show a regime or example where the need for adaptivity is shown (the ecology example does not demonstrate that for me but maybe I missed something?) and can you show an example which is used in the introduction from a long list of real world usecases where partial observability and online estimation is really needed and not only nice to have?

Experimentally I am just not sure it is demonstrated that the approach really solves the problems which require online estimation for example? It would be convincing if the approach scales to a setting where amortized approaches like DAD would really not be applicable but with the nested particle filtering it might actually be that DAD is more scalable? If that is true then only adaptation would remain and for that it would be nice to have a usecase where static policies actually fail but I am not sure I see that usecase in the experiments? What do I miss?

**Limitations:**

The paper clearly motivates why they are focused on non amortized, fully online regime. And that regime is indeed interesting. Yet it would be interesting at the same time to see if there method actually handles that regime better than an amortized approach. It is somehow implied in the text due to scalability issues but that is not shown and it would be nice to have DAD as a baseline or another amortized model especially since they are already cited.

The experiments like the one in section 5.3 are relatively weak. I am not sure what I should read from the plot and why adaptivity and online estimation would be needed in that case. The ecology experiment rather seems to demonstrate that adaptivity is not needed? Please correct me.
In the best case there would be an example where the offline designs fail and that would really demonstrate the value of the work.

Just to make it clear, I would be willing to increase my score even if DAD or an amortized baseline outperforms the proposed approach for the demonstrated experiment sections but I believe it would be important to show this performance level and in the best case have an usecase where this would not be possible anymore. If that is not possible anymore I would likewise believe that is a valuable contribution.

**Strengths And Weaknesses:**

The paper is nice to read, the plots are very clean and clearly readable. in addition, the paper addresses an important problem of experimental design for dynamical systems. The key aspect or novelty is that the approach can handle partial observability and online design at the same time. This is especially interesting since most recent works in machine learning rather focus on the amortized setting. However, why is it not explicitly compared to againts Foster et al works like DAD which should be applicable to the experiments shown in the experimental section. I can see why they should fail in a limit as motivated by the authors but I likewise do currently not see why their own proposed method would actually be more scalable or more applicable than DAD in many settings (there is of course a difference in the motivations but what I would argue is that there is a slight gap in the motivation in the introduction and the need for online decisions and the experiments which should be solvable by other methods too) So where precisely would DAD fail in the experiments in the experimental section? The limits with long horizon and all arguments are true in theory but the experimental section only shows settings where DAD would actually easily be trainable. Right now it is mostly implied but not shown or can you include at least one amortized baseline? Similarly why I agree with all theoretic motivations, can you demonstrate that the proposed solution is actually solving them?

Given the grand motivation in the introduction across nearly all disciplines ''The ability to guide data collection efficiently can accelerate learning in diverse domains, including material discovery (Lei et al., 2021; Lookman et al., 2019), DNA sequencing (Weilguny et al., 2023), sensor networks (Wu et al., 2023), and drug discovery (Lyu et al., 2019). ' the experimental section is fairly limited and the use cases potentially far away from the need which motivated the methods. E.g. Why and how would online design be required for the example from ecology in section 5.3? Does sequential adaptation really help here or is that really needed? From the plots I would actually argue that this might not be the case ...

---

> ### Author Rebuttal · Authors · 2026-03-30
>
> Thank you for the thoughtful and constructive review. We appreciate your positive assessment of the manuscript’s technical quality and presentation---especially that it is “technically very good and nice to read,” with “plots [that] are very clean and clearly readable,” and that the key novelty is handling “partial observability and online design at the same time”. Below, we address your comments point by point.
>
> 1. **Amortised baseline.** We appreciate the suggestion to compare with amortised methods (e.g., DAD). However, **a direct comparison is not straightforward.** Our experiments are partially observable SSMs, where the key quantity for BED is the marginal likelihood $p(y_t\mid \theta,\xi_t)$, which is intractable due to latent-state marginalisation. Standard policy-based approaches typically assume a tractable likelihood, and their training objectives (EIG or bounds) rely on likelihood evaluation. Adapting DAD to our regime would therefore require non-trivial changes to the training objective/estimators, and **would be a new methodological contribution** rather than a drop-in baseline. We now clarify this more explicitly in the manuscript and highlight amortised BED for SSMs as possible future work.
>
> 2. **Scalability and trade-offs vs. amortised approaches (limitations).** We have now added a discussion of these trade-offs. Even if an **amortised method** were tailored to partially observed SSMs, it may still **struggle in long-horizon settings due to coverage/generalisation:** training must approximate a policy over an expanding history space, typically requiring many long rollouts of latent trajectories and repeated belief updates. For large state spaces (e.g., moving-source trajectories that diverge substantially), adequate coverage becomes prohibitive. The step-DAD line of work (Hedman et al., 2025) explicitly motivates intermediate retraining to reduce suboptimality, indicating that **one-shot amortisation may not remain optimal as experiments progress.**
> This does not mean our approach is universally better: **amortised methods can be very strong and fast for short-to-moderate horizons, whereas BAD-PODS trades training-free deployment for higher per-step computation.** As it relies on nested PF, it also inherits particle-method challenges, with required particles increasing rapidly with state/parameter dimension.
>
> 3. **What we mean by adaptivity and where it is needed.** We clarify in the manuscript that **adaptivity refers to sequential BED**: we choose $\xi_t$ using the updated posterior $p(\theta,x_{0:t-1}\mid h_{t-1})$, rather than fixing $\xi_{1:T}$ in advance. This does not require the design to vary dramatically over time. **Sequential BED is adaptive because it conditions on realised data and can correct earlier suboptimal choices as information accumulates.**
>
>     Separately, **the online/recursive aspect in our work refers to maintaining inference and design updates without replaying the full history, i.e., with a linear-time recursion.** This matters in long-horizon partially observed settings (e.g., epidemiological surveillance or ecological monitoring that can last months/years). Even with time between measurements, repeatedly reprocessing the full history yields runtime and memory costs that grow with $T$ and eventually become limiting.
>
> 4. **Why include the ecological growth experiment.**
> **This experiment is meant to validate the more general setting where the design affects *both* the transition and observation equations, which makes EIG/gradient computation more challenging.** Even when the optimal design is approximately stationary, sequential inference remains essential because the latent state is unobserved and trajectories can diverge across realisations. We revised the experimental section to make this clearer.
>
> 5. **Failure in static BED and amortised approaches.**
> We added a clearer discussion of distinct failure modes. **Static BED** optimises $\xi_{1:T}$ offline from priors, so it **can fail at long horizons because (1) the optimisation becomes high-dimensional/non-convex and (2) cannot correct mistakes after seeing data**. In our experiments, this appears as underperformance and infeasibility at longer horizons due to memory growth. **Amortised methods can fail due to insufficient training coverage over histories/trajectories in large state spaces,** motivating retraining during deployment (as in step-DAD). **A direct empirical demonstration of amortised failure in our exact setting is not straightforward**, since adapting existing amortised methods to latent-state marginalisation and our objective is itself non-trivial.
>
> Once again, thank you for the thoughtful comments. We believe the revisions address the issues you raised, and we would be glad if you could take into account these changes when reconsidering the score. **We are eager to provide additional clarifications if any concerns still remain.**

---

> > ### Author Rebuttal · Reviewer_q42P · 2026-04-02
> >
> > Thanks a lot for the rebuttal. Indeed I think the motivation and importance of the work is high and that it aims to address a fundamental problem. Yet I am not sure the experiment section lives up to the motivation in the introduction. I fully agree what you write there and really hope you keep pushing in hat direction but I am not (yet) convinced you show a convincing solution for that in the experiments.
> >
> > Since the original DAD there seem to have been advances on non-likelihood based methods e.g. [1]. Why would that not work or be applicable here? Regarding the discussion with amortized methods, this seems quite inconclusive. They may struggle with coverage but they might likewise not struggle with coverage as is the case in other applications. For the examples provided in the experiment sections it is at least hard to imagine that this would be a limit. At least I would have hoped to see an exploration of the limits of the proposed approach in the experiment section minimally with an example which pushes scalability and performance in the experiment sections rather than just offering what I would see as of now as proof of concepts.
> >
> >
> > [1] Implicit Deep Adaptive Design: Policy-Based Experimental Design without Likelihoods

---

> > > ### Author Response · Authors · 2026-04-06
> > >
> > > Thank you for engaging and pushing on these points. We mention a broad class of applications in the introduction to motivate the problem setting. As it is a complex problem, we view our method---which addresses joint online design optimisation and partial observability, with consistent EIG estimators---just as an enabling step toward those applications. Then, we use illustrative experiments to validate the methodology, rather than to demonstrate end-to-end performance in a specific deployed domain. We have rephrased the manuscript to make this point clearer.
> > >
> > > Regarding iDAD, we agree it is among the closest relevant methods because it targets likelihood-free problems. However, a direct comparison is not straightforward for both *setting* and *objective* reasons. Our focus is **fully online** operation, whereas iDAD relies on offline training over simulated histories. Even if deployment is fast, the resulting design policy is learned offline and, at test time, decisions are driven by prior information and simulated training data rather than being optimised on-the-fly from the actual data collected. This would explain the suboptimality of policy-based methods observed in Hedman et al. (2025). Thus, we are considering different operational settings/assumptions. In addition, iDAD optimises an approximate lower bound (sPCE) rather than a consistent estimate of the EIG; therefore, comparing sPCE values to EIG values may not be meaningful. For these reasons, while iDAD is relevant, we believe the underlying assumptions differ enough that ensuring a fair comparison is non-trivial.
> > >
> > > Finally, we agree that amortised methods may or may not suffer from coverage issues depending on the application. Our point is not that they *must* fail in the examples included in the manuscript, but that amortised and non-amortised regimes make different assumptions and target different operational constraints. In particular, our setting prioritises fully online operation (no offline training) and accepts higher per-step computational cost in order to eliminate approximation errors, since we obtain theoretically consistent estimators (rather than prioritising fast deployment). This can be important in settings where trust is needed in estimates of the EIG because, for instance, selecting a new design implies a costly experiment like a randomised control trial.
> > >
> > > Hedman et al. (2025). Step-DAD: Semi-amortized policy-based Bayesian experimental design. arXiv preprint arXiv:2507.14057.

---

### Official Review · Reviewer_e51L · 2026-03-10

**Soundness:** 3
**Presentation:** 3
**Significance:** 4
**Originality:** 3
**Overall Recommendation:** 5
**Confidence:** 4

**Summary:**

This paper considers online Bayesian experimental design in partially observed nonlinear dynamical systems modeled via a state-space representation. It proposes BAD-PODS, a method that uses nested particle filters together with online expected-information-gain optimization to select designs sequentially while jointly inferring latent states and unknown parameters. The main technical contribution is a latent-state-marginalized estimator of expected information gain and its gradient, along with an asymptotic consistency result for the estimator. Empirically, the method outperforms random and static baselines on several simulated tasks and approaches the performance of an oracle grid-search variant.

**Compliance With Llm Reviewing Policy:**

Affirmed.

**Final Justification:**

The rebuttal adequately addressed my concerns so I am keeping my recommendation as accept. The paper is interesting as it tackles online experimental design for nonlinear dynamical systems when the states are only indirectly observed.

**Key Questions For Authors:**

1. The paper provides a consistency result for the EIG estimator under regularity assumptions. How sensitive is the method in practice when these assumptions are only approximately satisfied in harder nonlinear state-space models?
2. The empirical comparison is mainly against random design, static BED, and an oracle grid-search variant of the same approach. Could the authors compare against, or at least discuss, stronger external baselines for online design in partially observed settings?
3. Since the main evaluation metric is expected information gain, could the authors also report more direct downstream metrics such as parameter estimation error, state estimation error, or posterior calibration across all experiments?

**Limitations:**

It would help to state more clearly that the consistency result applies under regularity assumptions and that performance depends on particle counts and optimization settings.

**Strengths And Weaknesses:**

The paper studies an important and nontrivial problem, namely online Bayesian experimental design (BED) for partially observed dynamical systems. This is a meaningful extension of standard BED, since the design must be optimized sequentially while inference proceeds through latent state dynamics rather than a tractable likelihood. The proposed method is well motivated for this setting. BAD-PODS combines nested particle filtering with online expected-information-gain optimization, and the latent-state-marginalized EIG and gradient estimators are the main technical contribution. The paper also presents a consistency result for the EIG estimator. Empirically, the evaluation is reasonably broad for a methods paper. The experiments cover three distinct nonlinear state-space models, and the moving-source example includes a more direct downstream metric via pointing error rather than only reporting information gain. Overall, the paper makes a credible case that the approach works in the setting it targets.

My main reservations are about how broadly the current theory and experiments support the paper’s practical claims. On the theory side, the paper provides an asymptotic consistency result for the EIG estimator under regularity assumptions. This is a meaningful guarantee, although it is established for a fairly regular regime, so it is less clear how much of that support extends to more challenging nonlinear models where filtering and design are most fragile. On the empirical side, the results are encouraging, but the comparisons are mainly to random design, static BED, and an oracle grid-search version of the same approach. That is enough to show the method is promising, but it leaves the case for broad practical superiority somewhat limited. I also think the evaluation relies heavily on expected information gain itself, so the benefit for downstream inference is sometimes more indirect than direct, although the pointing-error experiment helps address this. Overall, I found the paper technically sound and reasonably well justified, but the current evidence supports a somewhat narrower claim than the paper’s broad practical framing.

---

> ### Author Rebuttal · Authors · 2026-03-30
>
> We thank the reviewer for the careful reading and constructive assessment. We appreciate your view that the paper tackles an “important and nontrivial problem,” and that BAD-PODS is “well motivated”. We also appreciate your assessment that the evaluation is “reasonably broad for a methods paper,” covering “three distinct nonlinear state-space models,” and your overall conclusion that the paper is “technically sound and reasonably well justified,” making a “credible case that the approach works in the setting it targets.”
> We address your comments below.
>
>  1. **Sensitivity to assumptions.**
> We now clarify this in the revised manuscript. In practice, instability typically appears first in the EIG integrand $\log(L/Z)$, rather than the filtering recursion. If the observation model yields very small or numerically zero likelihood values for a non-negligible set of samples---e.g., near-deterministic observations, bounded-support/truncated likelihoods, many outliers, or model mismatch---then $L, Z$ can become extremely small, making $\log(L/Z)$ unstable (high-variance gradients or occasional NaNs). Increasing $M,N$ reduces Monte Carlo variance, but does not fully fix regimes where the log-ratio is intrinsically ill-conditioned. **Thus, in practice, mild/occasional deviations (e.g., rare outliers) typically manifest as increased estimator variance and can often be mitigated by larger $M,N$ and numerical stabilisation.** However, repeated violation of the positivity/boundedness condition (Assumption C5), can break consistency. More generally, **violations of the stability/regularity (Assumption C4) can also degrade performance by making filtering highly sensitive to $\theta$, increasing estimator variance and slowing convergence.**
>
>  2. **Amortised methods.**
> We expanded the discussion of trade-offs with amortised methods. Many mortised methods (e.g., DAD) assume tractable likelihoods, so their training objectives rely on likelihood evaluations. **Adapting them to partially observed SSMs would therefore require non-trivial changes to objectives/estimators, becoming a new methodological contribution rather than a baseline.** Even if adapted, **amortised methods may struggle in long horizons due to coverage/generalisation:** training must approximate a policy over an expanding history space, and for large state spaces (e.g., trajectories that diverge substantially), adequate coverage becomes prohibitive. Step-DAD (Hedman et al., 2025) motivates intermediate retraining to reduce suboptimality, suggesting “one-shot” amortisation may not remain optimal as experiments progress. This does not mean our approach is always better: **amortised methods can be very strong/fast for short-to-moderate horizons, whereas BAD-PODS trades training-free deployment for higher per-step computation.** Since it relies on nested PF, particle requirements can grow rapidly with state/parameter dimension.
>
>  3. **Metrics beyond EIG.**
> We will add complementary metrics in the Appendix: (i) *parameter RNMSE* and (ii) *state RNMSE* (wrt. ground-truth). For the ecological growth model (tables below), the RNMSE trends match TEIG: BAD-PODS is close to the oracle and improves over random (static truncated when infeasible). We will include analogous results for SIR and moving-source as well.
>
> **State RNMSE** (bootstrap 95% CI):
> | t | Oracle | BAD-PODS | Random | Static |
> |---|---|---|---|---|
> | 5 | 0.029 [0.023,0.036] | 0.025 [0.019,0.034] | 0.035 [0.028,0.045] | 0.024 [0.019,0.030] |
> | 10 | 0.028 [0.022,0.034] | 0.027 [0.022,0.034] | 0.032 [0.025,0.041] | 0.029 [0.024,0.036] |
> | 15 | 0.023 [0.019,0.028] | 0.028 [0.022,0.036] | 0.032 [0.025,0.040] | - |
> | 20 | 0.024 [0.020,0.029] | 0.027 [0.021,0.036] | 0.032 [0.026,0.040] | - |
>
> **Parameter RNMSE** (bootstrap 95% CI):
> | t | Oracle | BAD-PODS | Random | Static |
> |---|---|---|---|---|
> | 5 | 0.494 [0.461,0.531] | 0.544 [0.498,0.591] | 0.558 [0.514,0.608] | 0.552 [0.508,0.597] |
> | 10 | 0.512 [0.456,0.571] | 0.545 [0.474,0.615] | 0.581 [0.529,0.637] | 0.533 [0.486,0.584] |
> | 15 | 0.488 [0.412,0.572] | 0.505 [0.408,0.613] | 0.606 [0.531,0.679] | - |
> | 20 | 0.464 [0.377,0.571] | 0.524 [0.421,0.636] | 0.613 [0.522,0.704] | - |
>
>  4. **Limitations.**
> We now state more clearly that consistency holds under regularity assumptions (boundedness/positivity and stability) and in the limit $M,N,L\to\infty$. In practice, performance depends strongly on the particle counts $(M,N)$, the jittering scale, and the optimisation budget $K$. We also highlight concrete regimes where assumptions may break---precisely where $\log(L/Z)$ becomes persistently ill-conditioned---which can degrade performance.
>
> Once again, thank you for the thoughtful comments. We hope the revisions address the issues you raised, and we would be glad if you could take into account these changes when reconsidering the score. **We are eager to provide additional clarifications if any concerns still remain.**

---

> > ### Author Rebuttal · Reviewer_e51L · 2026-04-02
> >
> > The rebuttal addressed my main concerns. It clarified the scope of the consistency result and explained more concretely what happens when the regularity assumptions are only approximately satisfied in practice. The authors also gave a reasonable response to my question about stronger amortized alternatives. The state and parameter RNMSE results help address my concern that the evaluation relied too heavily on EIG alone.

---

> > > ### Author Response · Authors · 2026-04-03
> > >
> > > Thank you for engaging and for the comments, which we think have improved the manuscript.

---

### Decision · Program_Chairs · 2026-04-30

**Decision:**

Accept (regular)

**Comment:**

This paper proposes BAD-PODS, a method for online Bayesian experimental design in partially observed nonlinear dynamical systems using nested particle filters and online expected-information-gain optimization. The idea is considered important and novel (e51L, foGd), technically sound with a consistent result (e51L, q42P), and well-motivated for sequential decision-making under partial observability (e51L, foGd). Although some weaknesses remain, including the presentation issues with missing conditioning variables and typos (t4FZ), given the overall assesment the decision is made with the expectation that the authors will thoroughly address all reviewer comments, particularly regarding presentation clarity, assumption verification, and experimental baselines, in the final version.